# Two zinc finger proteins with functions in m⁶A writing interact with HAKAI

Mi Zhang [1], Zsuzsanna Bodi [1], Katarzyna Mackinnon[2], Silin Zhong[1,3], Nathan Archer [4], Nigel P. Mongan [4,5], Gordon G. Simpson[2] & Rupert G. Fray [1✉]

The methyltransferase complex (m⁶A writer), which catalyzes the deposition of $N^6$-methyladenosine (m⁶A) in mRNAs, is highly conserved across most eukaryotic organisms, but its components and interactions between them are still far from fully understood. Here, using in vivo interaction proteomics, two HAKAI-interacting zinc finger proteins, HIZ1 and HIZ2, are discovered as components of the *Arabidopsis* m⁶A writer complex. HAKAI is required for the interaction between HIZ1 and MTA (mRNA adenosine methylase A). Whilst HIZ1 knockout plants have normal levels of m⁶A, plants in which it is overexpressed show reduced methylation and decreased lateral root formation. Mutant plants lacking HIZ2 are viable but have an 85% reduction in m⁶A abundance and show severe developmental defects. Our findings suggest that HIZ2 is likely the plant equivalent of ZC3H13 (Flacc) of the metazoan m⁶A-METTL Associated Complex.

[1] Plant Sciences Division, School of Biosciences, University of Nottingham, Sutton Bonington Campus, Loughborough LE12 5RD, UK. [2] School of Life Sciences, University of Dundee, Dow Street, Dundee DD1 5EH, UK. [3] State Key Laboratory of Agrobiotechnology, School of Life Sciences, The Chinese University of Hong Kong, Hong Kong, China. [4] School of Veterinary Medicine and Sciences, University of Nottingham, Sutton Bonington Campus, Loughborough LE12 5RD, UK. [5] Department of Pharmacology, Weill Cornell Medicine, New York, NY 10065, USA. ✉email: rupert.fray@nottingham.ac.uk

RNA modifications, which collectively constitute the epi-transcriptome, can facilitate posttranscriptional regulatory processes[1,2]. $N^6$-methyladenosine (m$^6$A) is necessary for the regulation of developmental and cell differentiation processes in most eukaryotes, and is one of the most studied internal modifications in eukaryotic messenger RNAs (mRNAs). It is deposited by the methyltransferase complex (m$^6$A writers) pre-dominantly at the 3′ end of transcripts[3,4]. The methylation can be removed by demethylases (m$^6$A erasers) and recognized by m$^6$A-binding proteins (m$^6$A readers). Identifying and character-izing the components of the m$^6$A writer complex is fundamental to understanding the regulatory roles of this modification.

The m$^6$A writer complex is formed by an assembly of proteins that are conserved between plants and animals. Studies on mammalian and *Drosophila* systems indicate that the writer complex can be separated into two sub-complexes. Methyl-transferase like 3 (METTL3, previously known as MT-A70) equivalent of the *Arabidopsis* mRNA adenosine methylase A (MTA)[5,6], and methyltransferase like 14 (METTL14, plant homolog mRNA adenosine methylase B [MTB])[7,8], constitute the m$^6$A-METTL Complex (MAC) and this joins with the m$^6$A-METTL Associated Complex (MACOM)[9]. MACOM is composed of Wilms' tumor 1-associating protein (WTAP, plant homolog FKBP12 interacting protein 37 [FIP37])[6,10], Vir-like m$^6$A methyltransferase associated (VIRMA, previously called KIAA1429, plant homolog VIRILIZER [VIR])[8,11], and HAKAI (also known as Casitas B-lineage lymphoma-transforming sequence-like protein 1 (CBLL1), plant homolog HAKAI)[8,11], and in animals it also contains and requires RNA binding motif protein 15 (RBM15) and the zinc finger CCCH domain-containing protein 13 (ZC3H13)[12,13]. The plant homolog of RBM15, FLOWERING LOCUS PA (FPA), which functions in flowering time control, was recently shown to co-purify with m$^6$A writer proteins but loss of this protein does not influence the global m$^6$A levels[14]. Despite homology searches, an equivalent of ZC3H13 has not been identified in plants[15]. In addition to the writer components, a negative regulator C$_2$H$_2$-type zinc finger protein 217 (ZFP217) also associates with the complex in mam-malian pluripotent stem cells[16].

We previously reported that *Arabidopsis* HAKAI is required for full m$^6$A methylation, with its knockout leading to a 35% reduction in m$^6$A abundance. Null mutations of any of the other reported m$^6$A writer proteins cause embryo lethality, and hypo-morphic partial knockouts lead to ~85% m$^6$A reduction and strong developmental defects[3,6,8]. Thus, HAKAI might be an accessary protein required for full m$^6$A methylation on all target transcripts or it might be responsible for m$^6$A deposition on just a specific subset of mRNAs. Here, we further characterize the function of *Arabidopsis* HAKAI and identify HAKAI-interacting zinc finger protein 1 and 2 (HIZ1 and HIZ2) as members of the m$^6$A writer complex. Our findings indicate that HIZ2 could be the long-sought plant homolog of ZC3H13, but in contrast to the other MACOM components, there is limited sequence con-servation with their presumed metazoan counterparts.

## Results

**HAKAI specifically interacts with two zinc finger proteins.** To facilitate the study of proteins interacting with HAKAI in vivo, HAKAI knockout mutant *hakai-2* was complemented with *HAKAI* under its own promoter and with a GFP tag at the car-boxy terminus, yielding the homozygous transgenic line—*HAKAIpro:HAKAI-GFP/hakai*. The GFP-tagged HAKAI protein was primarily localized to the nuclei in the primary root and lateral root initiation sites (Fig. 1a–c), in accordance with the localization of the main m$^6$A methylase MTA[6]. The m$^6$A level in

*HAKAIpro:HAKAI-GFP/hakai* was restored to that of WT (Fig. 1d), demonstrating that the GFP-tagged HAKAI protein acts normally and complements the *hakai-2* reduced m$^6$A phenotype.

We identified HAKAI interacting proteins by crosslinking of the nuclear fraction from *HAKAIpro:HAKAI-GFP/hakai* followed by anti-GFP co-immunoprecipitation and mass spectrometry[14]. Major proteins co-purifying with HAKAI were FIP37, VIR, MTA, two zinc finger proteins (AT1G32360 and AT5G53440, hereafter referred to as HAKAI-interacting zinc finger protein 1 and 2, HIZ1 and HIZ2, respectively) and a heat shock protein 70 family member (Hsp70-15, protein ID: A0A178W9Z4), whereas MTB was not among the directly-interacting proteins (Fig. 1e and Supplementary Data 1). To further investigate the role of HAKAI in the physical interaction between *Arabidopsis* m$^6$A writer complex members, we utilized the *MTApro:MTA-GFP/mta* line where the embryo lethal phenotype of *MTA* null mutant (Salk_114710) is complemented using the *MTA* coding sequence translationally fused to GFP at its carboxy terminus under the control of the *MTA* promoter (Supplementary Fig. 1a). As we were interested in finding out how HAKAI influences the composition of the writer complex, we crossed *hakai-2* with the *MTApro:MTA-GFP/mta* line and from the F$_3$ progeny we selected plants homozygous for both *hakai-2* and the *MTA* null mutant together with the complementing *MTApro:MTA-GFP* transgene (*hakai-2 MTApro:MTA-GFP/mta*). The localization and amount of GFP-tagged MTA protein in *hakai-2 MTApro:MTA-GFP/mta* was similar to *MTApro:MTA-GFP/mta* (Supplementary Fig. 1b, c). The m$^6$A level in *MTApro:MTA-GFP/mta* was at the WT level, and that in *hakai-2 MTApro:MTA-GFP/mta* was, as expected, similar to *hakai-2* measurements (Supplementary Fig. 1d), confirming the functional complementation by the GFP-tagged MTA under its own promoter.

We next carried out in vivo crosslinking and GFP pull-down experiments using both *hakai-2 MTApro:MTA-GFP/mta* and *MTApro:MTA-GFP/mta* lines. All known plant m$^6$A writer proteins, including VIR, FIP37, MTB and HAKAI, were found interacting with MTA (Fig. 1f and Supplementary Data 1). The two zinc finger proteins which were pulled out with HAKAI were also found as interacting partners with MTA (Fig. 1f). However, the knockout of HAKAI led to the disappearance of the interaction between MTA and HIZ1 whereas HIZ2 remained as an interacting partner with MTA (Fig. 1g, h). Therefore, we conclude that HAKAI is a bona fide m$^6$A writer protein in *Arabidopsis*, that HIZ1 and HIZ2 appear to be subunits of *Arabidopsis* m$^6$A writer complex, and that HAKAI is essential for the interaction between HIZ1 and MTA.

**HIZ1 levels are regulated by HAKAI, and HIZ1 ectopic expression reduces m$^6$A abundance.** Both HIZ1 and HIZ2 are annotated as zinc finger CCCH-domain containing proteins (TAIR database, https://www.arabidopsis.org/), however, their sequences share little homology. HIZ1 is composed of 384 amino acid residues (aa) while HIZ2 is more than three times as large (1181 aa or 1169 aa dependent on splice variant) (Supplementary Fig. 2). There are three zinc finger CCCH domains in HIZ1 protein sequence but only partial match is found in HIZ2 (Sup-plementary Fig. 2).

To address the function of HIZ1 in writing m$^6$A methylation, mutant *Arabidopsis* lines homozygous for gene-disruptive T-DNA insertions—Salk_045882 (*hiz1-1*) and Salk_000717 (*hiz1-2*) were identified (Fig. 2a). RT-qPCR analysis confirmed that *HIZ1* transcripts were absent in these two lines (Fig. 2b), but they displayed WT phenotypes, indicating that the loss-of-function of HIZ1 has negligible impact on overall growth and development. Recombinant constructs encoding carboxy terminus GFP-tagged

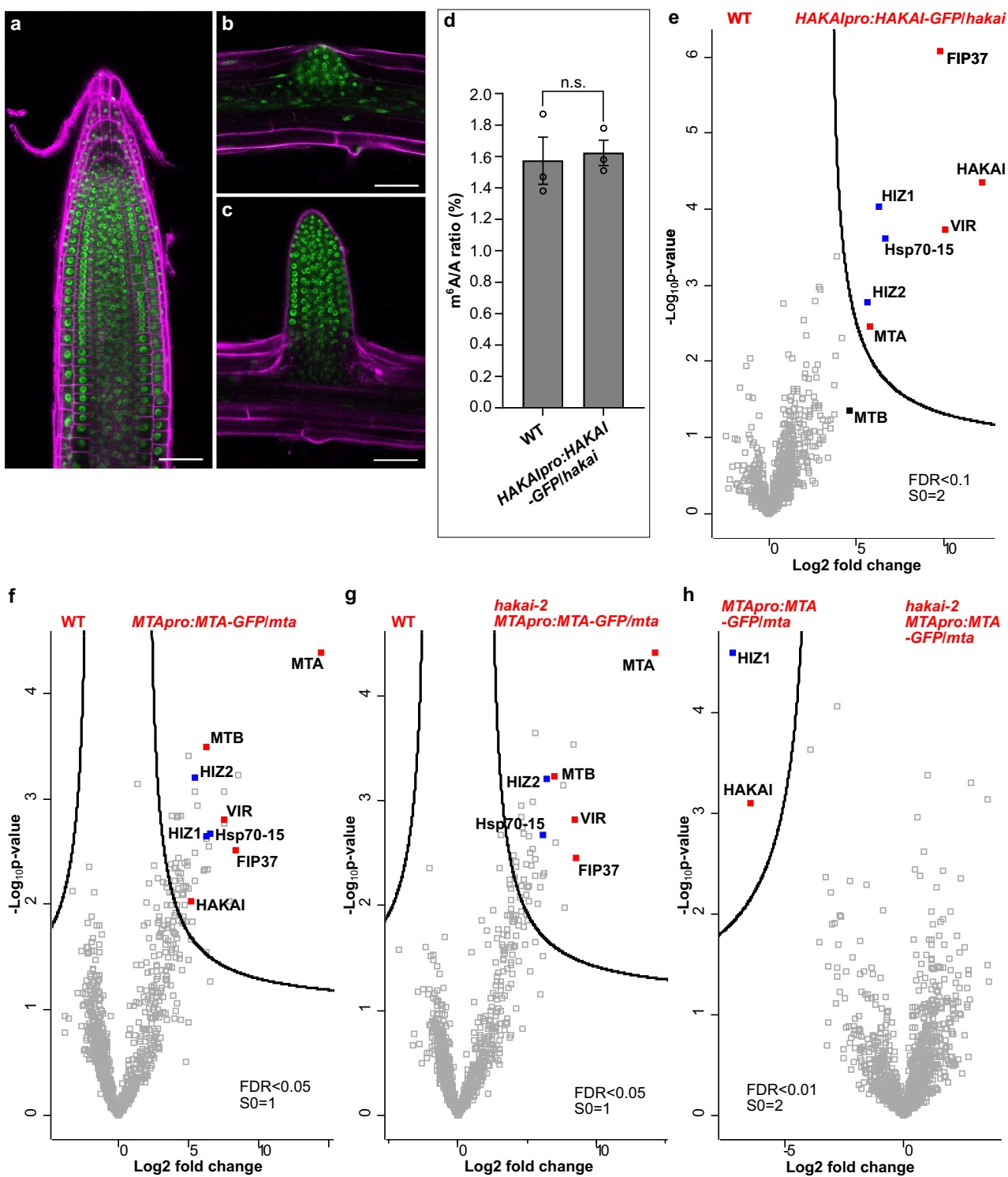

HIZ1 either under its own promoter or the constitutive cauliflower mosaic virus (CaMV) *35S* promoter, were constructed and transformed into *hiz1-1* and wild-type (WT) respectively to generate a complementation line (*HIZ1pro:HIZ1-GFP/hiz1*) and an ectopic overexpressor line (*35Spro:HIZ1-GFP/WT*) (Supplementary Fig. 3a). Assays on 2-week-old seedlings showed that the m$^6$A amount in *hiz1* mutants and the complementation line stayed unchanged relative to WT, while that in the overexpressor lines driven by *35S* promoter decreased by 24% (Fig. 2c). When

the *HIZ1* promoter rather than the constitutive *35S* promoter was driving *HIZ1-GFP* expression (*HIZ1pro:HIZ1-GFP/WT*), the m$^6$A to A ratio remained at the WT level (Fig. 2c). As expected, the expression of *35Spro:HIZ1-GFP* was observed mainly in the nuclei and was strongly expressed along the whole root (Supplementary Figs. 3b and 4). The localization of GFP-tagged HIZ1 under its own promoter was also nuclear, but within root structures, it was found predominantly in cells at the root tips and in root hairs (Fig. 2d, e and Supplementary Fig. 4). In the root

**Fig. 1 Characterization of the *HAKAIpro:HAKAI-GFP/hakai* line and in vivo analysis of proteins interacting with HAKAI or MTA using GFP-tagged baits. a–c** The localization of GFP-tagged HAKAI in roots of 5-day-old *HAKAIpro:HAKAI-GFP/hakai*. **a** A primary root tip. **b** A lateral root initiation site. **c** A formed lateral root. Scale bar = 50 μm. Experiments in **a–c** were repeated independently at least three times, and representative images are shown. **d** m[6]A levels checked by two-dimensional thin layer chromatography (TLC) analysis. Data represent mean ± SE from three biological replicates and statistically significant differences relative to WT were analyzed by two-sided unpaired *t*-test (n.s. no significance). *p* = 0.7845. **e–h** Volcano plots showing proteins co-purified with HAKAI or MTA. The distribution of quantified proteins was plotted according to the $Log_2$ fold change of label free quantification (LFQ) intensities and $-Log_{10}$p-value obtained from two-sided student *t*-test of three independent experiments. Significantly enriched proteins were separated from others by a hyperbolic curve. Proteins indicated with filled squares represent known m[6]A writer complex members (red) and additional components identified in this work (blue). **e** *HAKAIpro:HAKAI-GFP/hakai* versus WT. HIZ1 and HIZ2 refer to HAKAI-interacting zinc finger protein 1 and 2 (AT1G32360 and AT5G53440, respectively). MTB is indicated not directly interacting with HAKAI, labeled with a black filled square. **f** *MTApro:MTA-GFP/mta* versus WT. **g** *hakai-2 MTApro:MTA-GFP/mta* versus WT. **h** *hakai-2 MTApro:MTA-GFP/mta* versus *MTApro:MTA-GFP/mta*. Source data are provided as a Source Data file.

tips, HIZ1 localized to the columella and stem cell niche (including the quiescent center and stem cells surrounding it) of the primary roots (Fig. 2d, e). This stem cell niche localization is somewhat different from that of HAKAI and FIP37, which show reduced expression in these cells and higher expression in the apical meristem and basal meristem zones of the primary root tip where HIZ1-GFP is lower (Fig. 2d). The localization of GFP-tagged HIZ1 in *HIZ1pro:HIZ1-GFP*/WT was similar but the protein amount was slightly higher (Supplementary Fig. 3b, c). During lateral root development, the expression of *HIZ1pro:-HIZ1-GFP* was maintained at a low level prior to lateral root emergence from the primary root, and afterwards its expression increased and concentrated in the lateral root tips (Supplementary Fig. 3d). In contrast, the expression of GFP-tagged HAKAI or FIP37 was much stronger during lateral root initiation and development (Supplementary Fig. 3d).

The proteomics data suggest that HAKAI has an effect on the interaction between MTA and HIZ1. Therefore, *HIZ1pro:HIZ1-GFP/hiz1* was crossed with *hakai-2* to check whether the expression and localization of GFP-tagged HIZ1 would change in a *hakai-2* background. The homozygous line, *hakai-2 HIZ1pro:HIZ1-GFP/hiz1*, was used for imaging the localization of GFP-tagged HIZ1. We found that the disruption of HAKAI led to an increase of *HIZ1pro:HIZ1-GFP* expression in the primary root tip (Fig. 2e). We observed a similar increase in expression in the lateral roots of *hakai-2 HIZ1pro:HIZ1-GFP/hiz1*, and this was noticeable before lateral root emergence (Supplementary Fig. 3d). Western blotting confirmed the higher level of GFP-tagged HIZ1 upon the knockout of HAKAI (approximately a 3-fold increase) (Fig. 2f). To test whether this increase of protein level is due to an increase in *HIZ1* transcript levels, we performed RT-qPCR to check the *HIZ1pro:HIZ1-GFP* transcript abundance in *HIZ1pro:-HIZ1-GFP/hiz1* and *hakai-2 HIZ1pro:HIZ1-GFP/hiz1*. The transcript levels for *HIZ1pro:HIZ1-GFP* were 2.8-fold increased in the absence of HAKAI (Fig. 2g). Thus, HAKAI does not seem to act as a posttranslational regulator of HIZ1, but rather affects promoter activity or transcript stability.

**Reducing m[6]A modulates auxin responses and root development**. Plants with decreased m[6]A levels show reduced apical dominance and partial agravitropic root growth as a result of impaired auxin responses[3,6,17]. Although the root morphology of *hiz1-1* is similar to that of WT, the overexpressor line, *35Spro:-HIZ1-GFP*/WT-1, exhibited similar root phenotypes to known m[6]A writer mutants, including shorter primary roots and a decreased number of lateral roots (Supplementary Fig. 5).

Both primary and lateral root development and growth are controlled by the plant hormone, auxin, and an auxin gradient, established via polar transport, is critical for lateral root initiation[18]. Thus, in WT *Arabidopsis* plants, lateral roots can be induced along the entire length of the primary root by first

treating seedlings with *N*-1-naphthylphthalamic acid (NPA), which inhibits the major auxin efflux carriers[19], then treating with the synthetic auxin 1-naphthaleneacetic acid (NAA). Because NAA can freely enter cells by passive diffusion, its intracellular level is normally controlled by the efflux carriers[20]. Thus, the sequential treatment with NPA and NAA results in an auxin maxima and potential activation of all pericycle cells for lateral root initiation[21]. We wished to test if the low m[6]A lines were impaired in their ability to respond to this treatment. Four-day-old *Arabidopsis* seedlings were first placed on medium containing NPA for 3 days and then transferred onto medium containing NAA. On the 5th day of the NAA treatment, induced lateral roots were clearly visible along the whole primary root of WT plants (Fig. 3a). However, fewer induced lateral roots were observed in all mutants for m[6]A writer complex members apart from HIZ1 (Fig. 3a, b). This reduction was most severe in *vir-1* plants (Fig. 3a). In *mta* mutant plants, the reduction of lateral root number was restored to the WT levels by introducing the WT *MTA* transgene (Fig. 3a). We suspected that early cell division may be influenced by m[6]A deficiency. To further study the influence of m[6]A on early cell division in lateral root induction, the *mta ABI3:MTA* plant (where the *ABI3:MTA* construct complements the embryo lethal phenotype of the MTA knockout mutant but exhibits mutant phenotype in a mature plant)[3,6] was crossed with a mitotic marker line *CycB1;1::GUS*[18], and the GUS expression was compared between the *MTA, CycB1;1::GUS* and *mta, CycB1;1::GUS* plants (*mta ABI3:MTA CycB1;1::GUS*). The GUS expression in the *mta ABI3:MTA* background was markedly decreased during lateral root induction compared with that in the WT background, indicating reduced initiation of cell division (required for lateral root initiation and development) (Fig. 3c). In addition, the GFP expression analysis of *MTApro:MTA-GFP/mta* treated with NPA and NAA demonstrated that the *MTApro:MTA-GFP* transgene was strongly expressed during the whole lateral root induction process, primarily at root tips and lateral root initiation sites (Supplementary Fig. 6). Altogether, this is consistent with a reduced auxin response in the *Arabidopsis* root upon m[6]A deficiency. When the same NPA and NAA treatments were performed on *hiz1* mutants and *HIZ1-GFP* lines (*HIZ1pro:HIZ1-GFP/hiz1, HIZ1pro:HIZ1-GFP*/WT and *35Spro:HIZ1-GFP*/WT-1), the overexpressor line where *HIZ1* was driven by the constitutive *35S* promoter showed decreased lateral root induction whilst other lines displayed WT levels of lateral root induction (Fig. 3b).

Whilst conducting these experiments, we observed that in all *HIZ1-GFP* lines (*HIZ1pro:HIZ1-GFP/hiz1, HIZ1pro:HIZ1-GFP*/WT and *35Spro:HIZ1-GFP*/WT-1), there were significantly increased root hair lengths and density following NPA treatment compared to WT (Fig. 4a–d). This morphological change was observed only in *HIZ1-GFP* lines (most significant in *HIZ1pro:HIZ1-GFP*/WT) and not in any other m[6]A writer mutants (Fig. 4a).

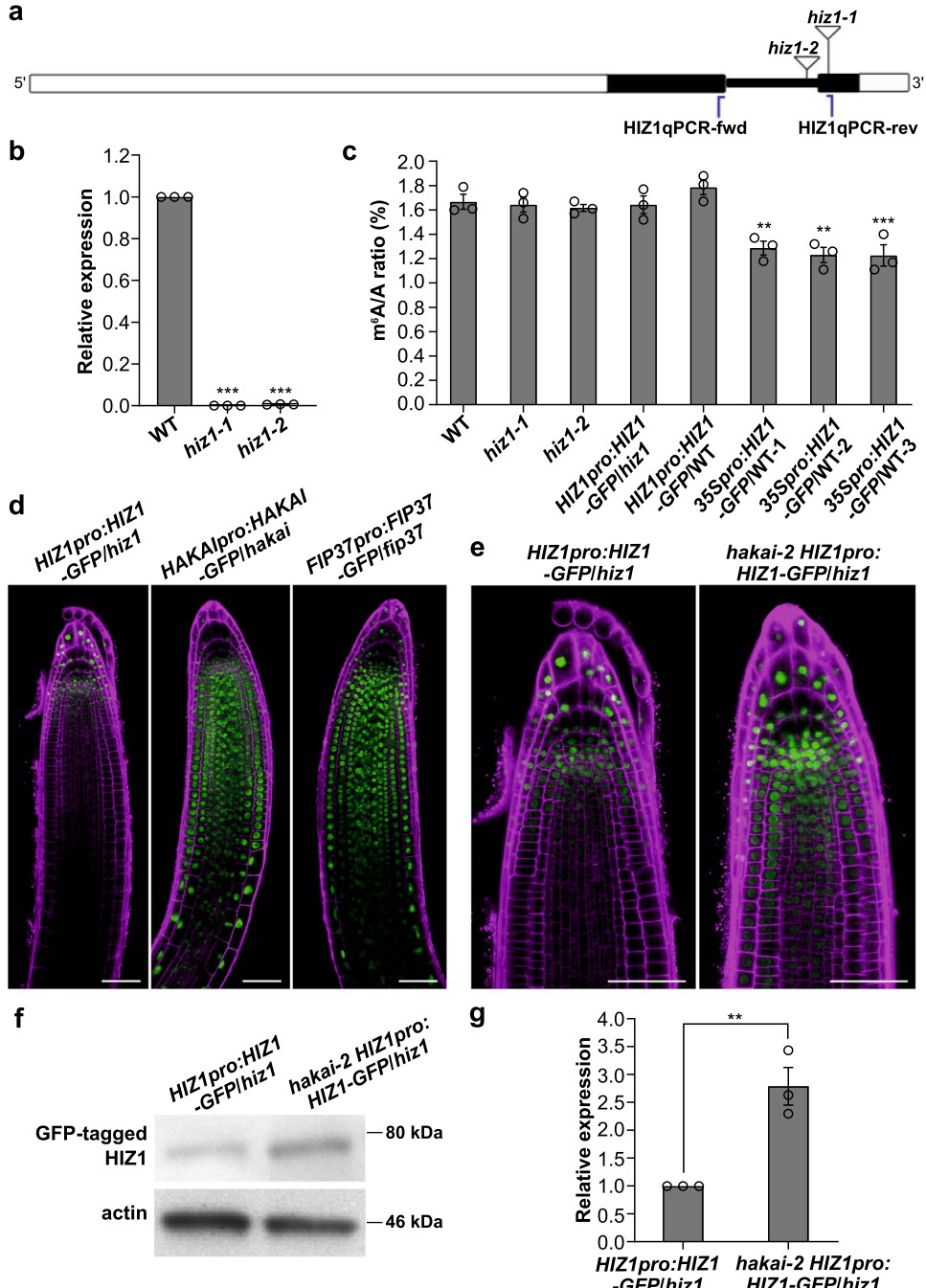

**Fig. 2 Characterizing *Arabidopsis* HIZ1. a** Schematic of *HIZ1* (AT1G32360) genomic DNA sequence with T-DNA insertion sites and the primer pair used for RT-qPCR. White rectangles denote UTRs of *HIZ1* genomic DNA, black rectangles denote exons and thick black lines represent introns. *hiz1-1* refers to Salk_045882 and *hiz1-2* refers to Salk_000717. Blue lines represent the locations of primers. **b** The transcript levels of *HIZ1* analyzed by RT-qPCR using the primer pair labeled in **a**. *CBP20* was used as a reference gene. Data here and in **c** represent mean ± SE from three biological replicates and statistically significant differences relative to WT were analyzed by One-Way ANOVA (one-sided test) and marked with asterisks (**$p < 0.01$; ***$p < 0.001$).
$p < 0.0001$ (*hiz1-1*); $p < 0.0001$ (*hiz1-2*). **c** m6A levels checked by two-dimensional thin layer chromatography (TLC) analysis. In one-way ANOVA test,
$p = 0.9996$ (*hiz1-1*); $p = 0.9907$ (*hiz1-2*); $p = 0.9996$ (*HIZ1pro:HIZ1-GFP/hiz1*); $p = 0.6393$ (*HIZ1pro:HIZ1-GFP/*WT); $p = 0.0036$ (*35Spro:HIZ1-GFP/*WT-1);
$p = 0.001$ (*35Spro:HIZ1-GFP/*WT-2); $p = 0.0009$ (*35Spro:HIZ1-GFP/*WT-3). **d**, **e** Localization of GFP-tagged proteins in primary root tips of 3-day-old seedlings. Scale bar = 50 μm. Experiments in **d**, **e** were repeated independently at least three times, and representative images are shown. **f** Western blot demonstrating the increased protein level of GFP-tagged HIZ1 in the *hakai-2* background. Experiments in **f** were repeated independently three times with similar results. **g** Transcript levels of *HIZ1pro:HIZ1-GFP* analyzed by RT-qPCR using the same primer pair as that in **b**. *CBP20* was used as a reference gene. Data represent mean ± SE from three biological replicates and the statistically significant difference was analyzed by two-sided unpaired *t*-test and marked with asterisks (**$p < 0.01$). $p = 0.0061$. Source data are provided as a Source Data file.

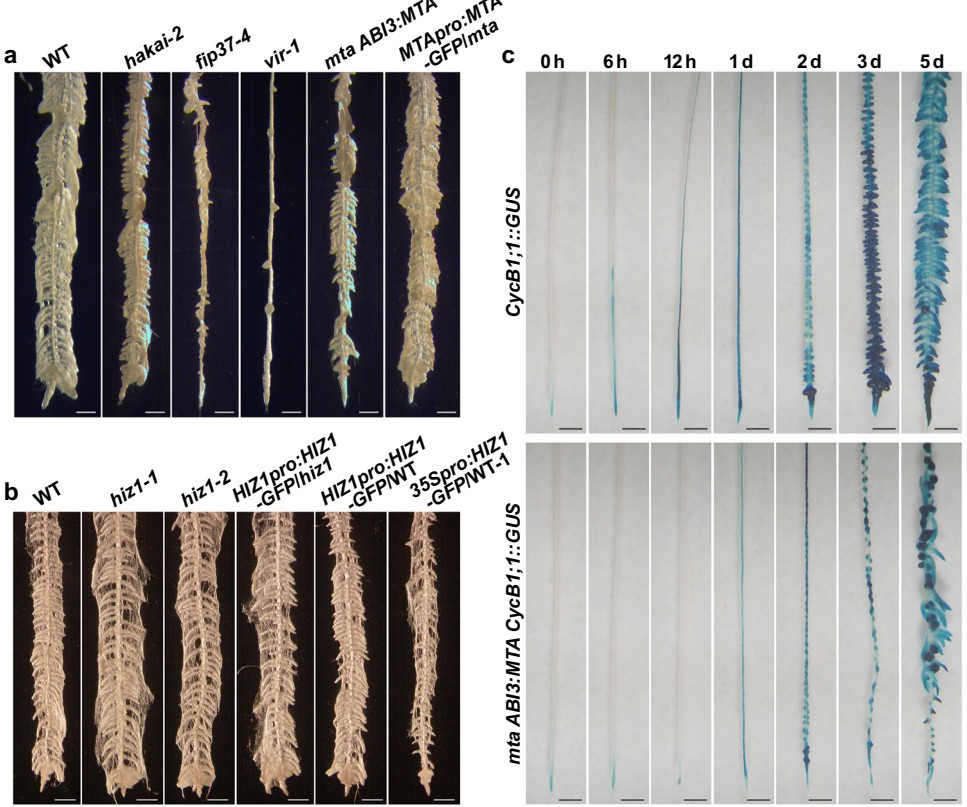

**Fig. 3 Root morphology under the pericycle activation for lateral root induction. a, b** Root morphology of seedlings after being treated with 1-naphthaleneacetic acid (NAA, 10 μM) for 5 days following *N*-1-naphthylphthalamic acid (NPA) treatment. Roots were soaked in 100% ethanol for a few seconds to minimize the effect of root hairs in photographing lateral roots. Scale bar = 500 μm. **c** GUS staining showing *CycB1;1::GUS* activities in WT and *mta ABI3:MTA* backgrounds during a time-course growth on NAA after NPA treatment. Scale bar = 500 μm.

*HIZ1pro:HIZ1-GFP* is expressed in hair cells and this expression is strongly elevated following NPA treatment. In contrast, *35Spro:HIZ1-GFP* is expressed in all root cells (including root hairs) (Supplementary Fig. 4). Consistent with the root hair phenotypes, *HIZ1-GFP* transgene abundance in *HIZ1pro:HIZ1-GFP/hiz1* increased by 2-fold relative to the WT *HIZ1* transcript level (Supplementary Fig. 7). This suggests that the *HIZ1-GFP* transgene not only complements the loss of *HIZ1* expression in *hiz1-1*, but also leads to the local overexpression of *HIZ1* to some extent, possibly due to the lack of regulatory sequences from the endogenous 3′ UTR (Supplementary Fig. 3a), which is itself methylated and is absent in the carboxy-terminal GFP tagged lines. In all of the *HIZ1-GFP* lines, the root hair physiological changes were also mirrored at the RNA level by the increased expression of the bHLH transcription factor *ROOT HAIR DEFECTIVE 6-LIKE 4* (*RSL4*) following NPA treatment (Fig. 4e). RSL4 is a key regulator of root hair genes, the constitutive expression of which promotes root hair formation[22]. The induction of *RSL4* is indicative of the initiation of root hair specific pathways in the NPA-treated *HIZ1-GFP* lines. However, this root hair promoting activity of HIZ1 appears to require the presence of HAKAI, as the positive effect on root hair development was lost when the *HIZ1-GFP* lines were crossed into genetic backgrounds in which HAKAI was knocked out (Supplementary Fig. 8).

**Transcriptome-wide analysis of m6A topology in WT, *hakai-2*, *fip37-4*, and *vir-1*.** HAKAI knockout lines have a 35% lower m6A/A ratio compared with WT. However, the m6A levels are higher than in hypomorphic mutants of other key complex members, and phenotypically *hakai* mutants resemble WT. The double mutants between *hakai* and core complex members exhibit a much-enhanced phenotype, characteristic of more severe m6A deficiency, than the single core complex mutants alone[8], indicating an additive or synergistic effect of the mutations. The reduced level of m6A seen in the *hakai* lines could be due to a general reduction of methylation across all target transcripts, or it could result from loss of methylation from just a specific subset. To address this question, we carried out methylated RNA immunoprecipitation with next-generation sequencing (MeRIP-Seq) on mRNAs extracted from 2-week-old seedlings of WT and writer mutants and compared results for *hakai-2* with those of WT and of the two other complex member mutants, *vir-1* and *fip37-4*. MeRIP-Seq enriches for m6A-containing mRNA fragments, which are defined as peaks relative to the input sample. We identified 12,715 peaks for WT, 13,546 for *hakai-2*, 8883 for *vir-1*, and 9809 for *fip37-4*. We did not expect peaks to disappear completely in *vir-1* and *fip37-4*, as both mutants are hypomorphic (~85% reduction in m6A), their complete knockout being lethal[6,8]. The topology and number of m6A peaks at the transcript level was very similar between *hakai-2* and WT, and significantly different in the *fip37-4* and *vir-1* mutants (Fig. 5a). All three mutants had a decrease in m6A methylation at the 3′ ends of transcripts (Fig. 5a), as well as a relative increase at the 5′ ends and across the coding sequence (CDS). Similar results for *fip37* were previously observed by Shen et al.[23]. At the genomic level, most of the m6A sites were found in protein coding transcripts and in long non-coding RNAs (lncRNAs) (Supplementary Fig. 9). There was an increase in m6A

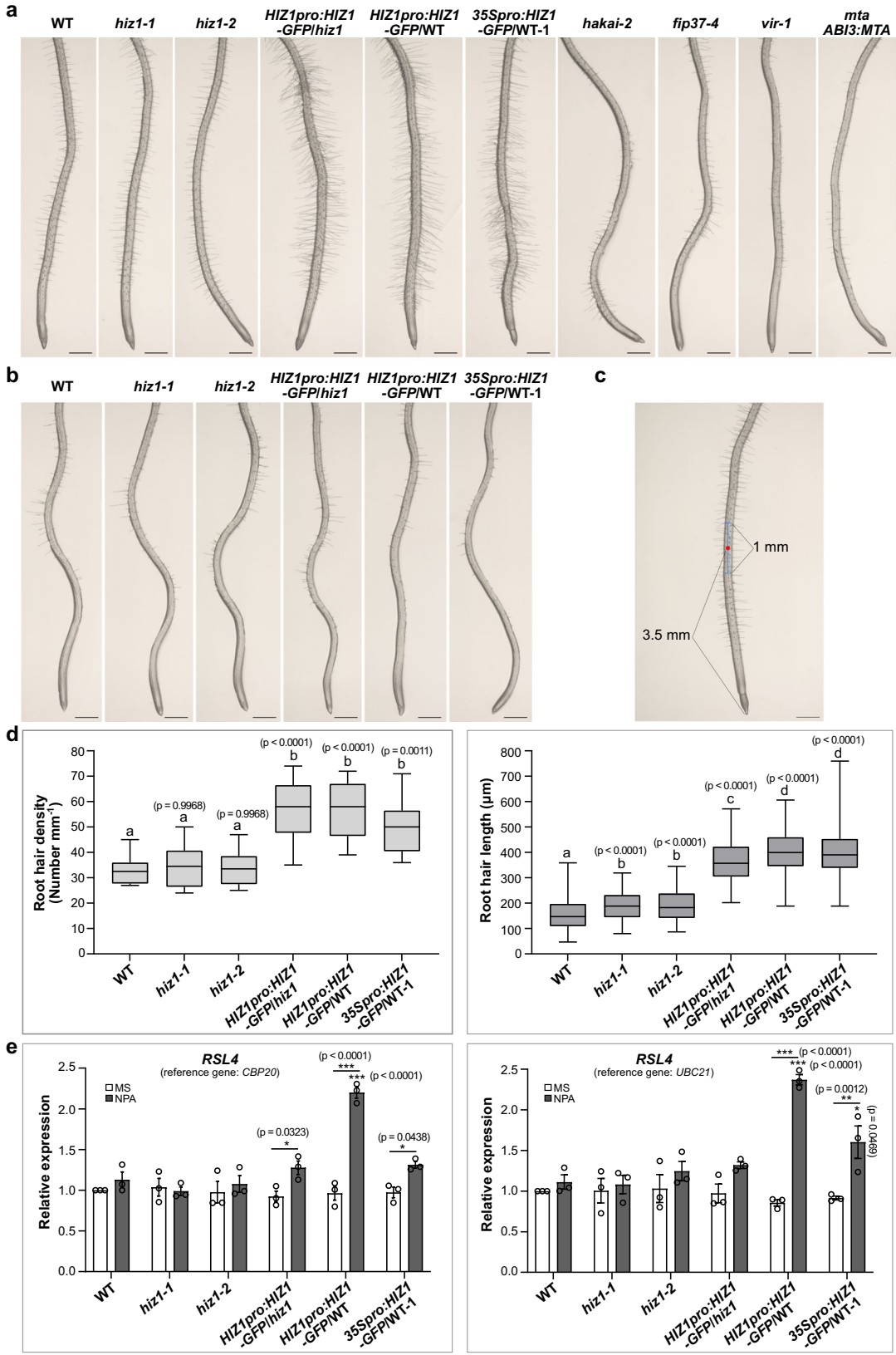

deposition in the intergenic regions for all three mutants, which was most pronounced in the *fip37-4* immunoprecipitation (IP) samples (Fig. 5b).

We wanted to test whether the topological differences in m$^6$A deposition can be biologically meaningful, globally. Thus, we carried out a GO enrichment analysis on the protein coding

methylated transcript lists which were present in WT and absent in *hakai-2* (798), *fip37-4* (1566) or *vir-1* (3079) (Supplementary Data 2–4). For *hakai-2*, we created a GOBP list containing those GO terms that were at least 2-fold enriched and had a *p* value <0.05 (Fisher test) (Supplementary Data 5). Some of the highest (50.2-fold) enriched processes were RNA localization to

**Fig. 4 Root hair morphology under the *N*-1-naphthylphthalamic acid treatment. a** Root morphology after being treated with *N*-1-naphthylphthalamic acid (NPA, 10 μM) for 6 days. Scale bar = 500 μm. **b** Roots from 0.5× MS control of the same age as those in **a**. Scale bar = 500 μm. Experiments in **a**, **b** were repeated independently at least three times, and representative images are shown. **c** A schematic illustrating how root hair density and lengths were measured. The number of root hairs in the middle 1 mm segment (shown using a blue segment with a red dot in the middle) of the imaged root represents the root hair density. The lengths of 10 root hairs on each side of the primary root in the middle 1 mm segment were measured to indicate its representative root hair length. Scale bar = 500 μm. **d** Statistical data showing root hair density and lengths. Ten seedlings were used for the measurement for each line. Data represent mean ± SE and statistically significant differences were analyzed by One-Way ANOVA (one-sided test). *p* values labeled in the diagram represent the comparisons with WT and letters represent significance at *p* < 0.01. In box plots, the center line in each box indicates the median. The lower and upper bounds of each box represent the first quartile (25%) and the third quartile (75%), respectively. The bottom and top of whiskers denote the minimum and maximum, respectively. **e** Transcript levels of *RSL4* analyzed by RT-qPCR. Samples were roots grown on NPA or the control (0.5× MS medium without NPA) for 6 days after germinating on 0.5× MS for 4 days. Data represent mean ± SE from three biological replicates. Statistically significant differences were analyzed by two-way ANOVA (one-sided test) and marked with asterisks (**p* < 0.05; ****p* < 0.001). Asterisks labeled just above the column are for comparing with WT grown under the same condition and those labeled above both columns for each line are for the comparison between two growth conditions (with or without NPA). Source data are provided as a Source Data file.

chromatin (GO: 1990280), regulation of antisense RNA transcription (GO: 0060194), circadian regulation of translation (GO: 0097167) and regulation of transcription by RNA polymerase V (GO: 1904279). We were particularly interested in processes that are involved in root hair initiation or elongation, as our data showed that HIZ1 is involved in root hair development in a HAKAI-dependent manner. We identified six transcripts that were 2.49-fold enriched in association with root hair cell differentiation (GO: 0048765), trichoblast maturation (GO: 0048764), and 2.41-fold enriched in trichoblast differentiation (GO: 0010054) in WT versus *hakai-2* mutant (Supplementary Data 5). All of these six transcripts were methylated at their 3′ ends in WT plants, and this was missing from the *hakai-2* samples. MeRIP-Seq profiles of actin 2 (*ACT2*, AT3G18780) and *Arabidopsis* Rac-like GTP binding protein 5 (*ARAC5*, AT1G75840) belonging to this GO term enriched set are presented in Fig. 5c. The 1566 differential peaks in WT versus *fip37-4* mutant and 3079 for WT versus *vir-1* were also tested for GOBP enrichment. We expected terms associated with embryo development and root development to be represented as we already have phenotypic evidences for these processes being influenced by the lack of mRNA methylation[6,8]. For *fip37-4*, one of the top categories was embryo development (GO: 0009790) (2.06-fold enrichment), as well as root development (GO: 0048364) (2-fold enrichment) (Supplementary Data 5). The embryo development was only 1.64-fold enriched in the *vir-1* samples, however, lateral root formation (GO: 0010311) was enriched 3.19-fold (Supplementary Data 5). The root hair development associated transcripts were also enriched in *fip37-4*, however different sets of transcripts are losing methylation in this mutant compared to *hakai-2*. One of the highest enriched terms in the *fip37-4* list was mRNA polyadenylation (GO: 0006378), with a 6.21-fold enrichment (Supplementary Data 5). Association between mRNA methylation and polyadenylation site choice was already established by Parker et al. in the *vir-1* mutant line[24]. We found that all six members of the group including nuclear poly(A) polymerase 4 (*PAPS4*, AT4G25550) and *CFIS2* (AT4G25550) are missing their m6A peaks in the 3′ UTR (Fig. 5d) or significantly reduced (*FY*, AT5G13480) in the *fip37-4* mutant.

We also wanted to see if there was any connection between the methylation levels of any transcript and their abundance. We carried out a transcriptomic analysis of all three mutants where we identified differentially expressed genes (1.5-fold change cut off). We then looked at the proportional representation of the transcripts that lost their methylation in these gene lists. We did not find any over representation of lack of methylation in the up or down regulated sets in any of the mutants (as an example, *vir-1* analysis is in Supplementary Data 6). Next, we used the dataset from Lavenus et al.[25], where the authors specifically identified genes that are regulated by auxin response factor 7 (ARF7), a transcription factor which is required together with ARF19 for lateral root initiation in response to auxin. We extracted all of the methylated genes from this dataset, which gave us a list of 90 m6A methylated transcripts, more than half (52%) of the verified ARF7 responsive gene dataset (173 genes) (Supplementary Data 7). We again wanted to know whether the loss of methylation in ARF7 responsive transcripts would affect their expression at the transcriptional level. Therefore, we extracted those genes that lose methylation in the *vir-1* mutant (a list of 19 genes, Supplementary Data 7). Out of the 19 transcripts that lost methylation, only two were found in the *vir-1* differentially expressed gene list. Nevertheless, these two transcripts *ARF19* and auxin resistant 1 (*AUX1*) are both instrumental in lateral root development and have decreased expression in *vir-1*. This finding is in agreement with defective lateral root development phenotypes we observe in m6A deficient plants[8]. Next we looked at how methylation affects gene expression during root hair development. From a gene expression dataset of Lan et al.[26] that reports a root hair dataset compared with data obtained from analyzing all root tissues, we identified 365 methylated transcripts out of 635 upregulated genes. This equates to a 57% representation (representation 1, *p* < 0.152) of the mRNA methylation in those transcript that increase abundance during root hair development (Supplementary Data 8). Next, we looked at the representation of methylated transcripts in the down-regulated list of genes, which was 72.5%, a statistically significant increase (representation 1.3, *p* < 1.042e−29, http://nemates.org/MA/progs/overlap_stats.html) (Supplementary Data 8). This suggests that during root hair development, the presence of m6A is associated with a decrease in abundance of the relevant transcripts.

**HIZ2 is required for full m6A methylation and normal growth and development**. From the co-immunoprecipitation experiments we identified two zinc finger proteins (HIZ1 and HIZ2) as HAKAI interacting partners. While HIZ1 is a writer complex member only by its interaction with HAKAI, HIZ2 was shown to stay bound to MTA even when HAKAI was absent. Thus, we wanted to know whether HIZ2 is required for m6A deposition. We identified three homozygous T-DNA insertion mutants disrupting the expression of *HIZ2*, referred to as *hiz2-1* (Salk_120590), *hiz2-2* (Salk_020625), and *hiz2-3* (Salk_126486) (Fig. 6a). RT-qPCR analysis confirmed that the correct transcripts of *HIZ2* in *hiz2-1* and *hiz2-2* were absent and in *hiz2-3* they were markedly reduced and truncated (Fig. 6b, c). Unlike the major hypomorphic m6A writer mutants (*mta ABI3:MTA*, *fip37-4* and *vir-1*) where a full knockout appears to be lethal[3,6,8], the three *hiz2* mutants are viable but demonstrate

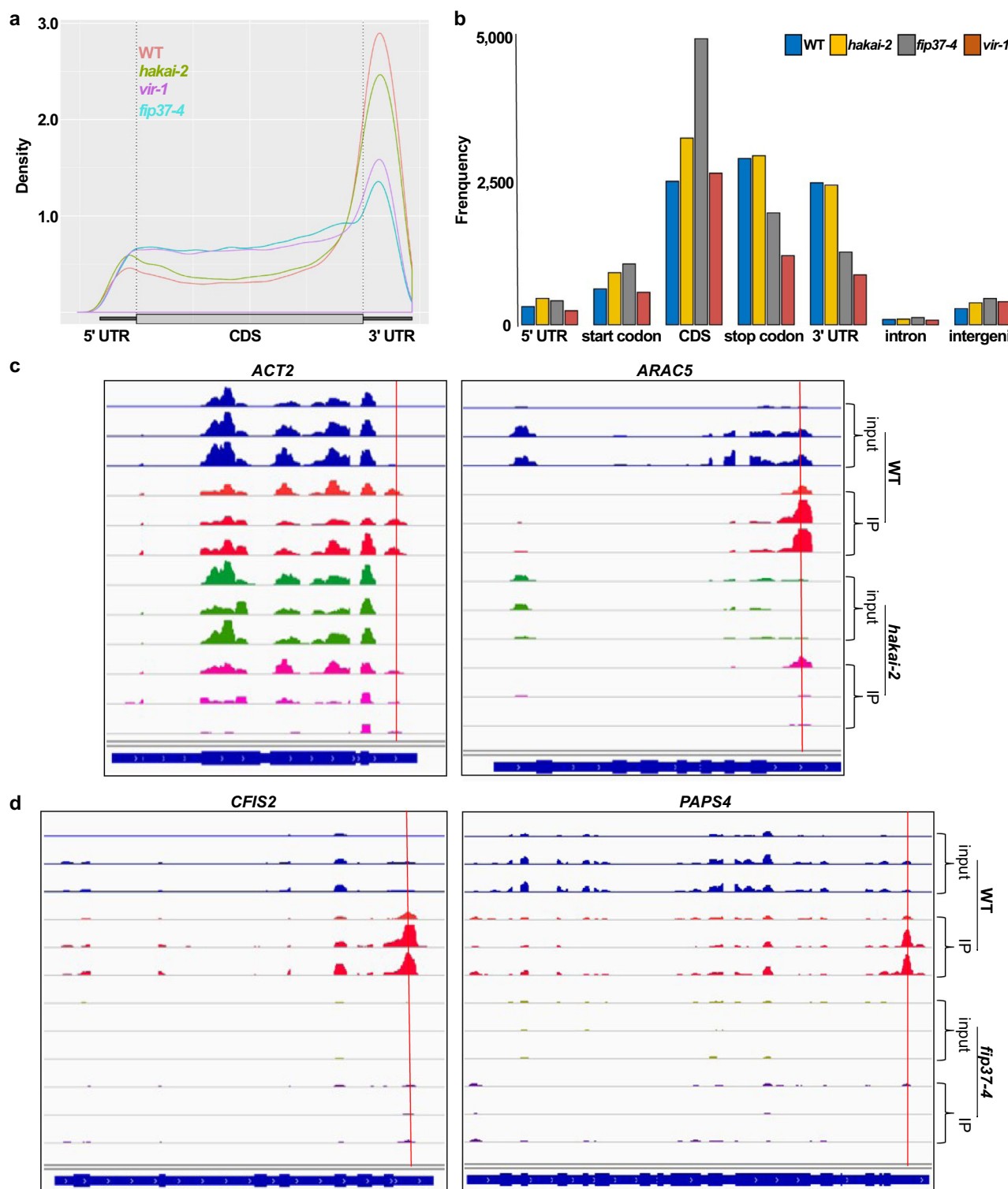

**Fig. 5 MeRIP-Seq analysis of three key components of the m⁶A writer complex mutants. a** Metagene profile of the global distribution of m⁶A in WT, *hakai-2*, *fip37-4* and *vir-1*. **b** Frequency of m⁶A sites in the different transcript parts in relation to their genomic position. **c** m⁶A topology differences between *hakai-2* and WT in two key transcripts involved in root hair development (images from Integrative Genomics Viewer [IGV]). IP: immunoprecipitation. *y* axis scale for *ACT2*: 0–4000 reads; *y* axis scale for *ARAC5* inputs: 0–1000 reads and for IPs: 0–4000 reads. Red lines here and in **d** represent the positions of predicted m⁶A peaks. **d** m⁶A topology differences between *fip37-4* and WT in two examples of the GOBP mRNA polyadenylation group (images from IGV). *y* axis scale for both *CFIS2* and *PAPS4*: 0–1000 reads.

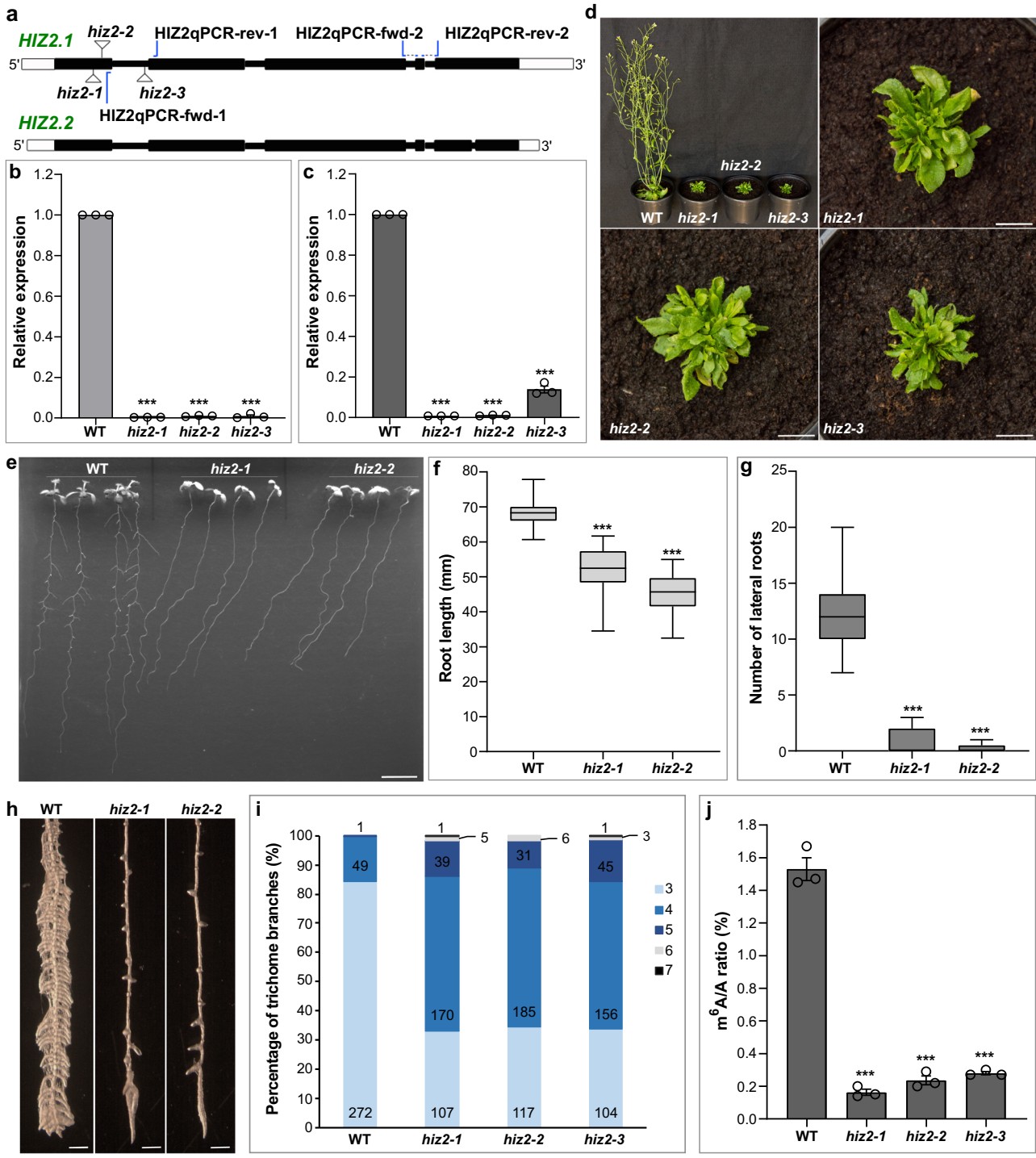

strong developmental defects, including retarded growth, reduced apical dominance and small and bushy rosette leaves, which are similar to, but more severe than, the phenotypes of the major hypomorphic m⁶A writer mutants (Fig. 6d and Supplementary Fig. 10). In addition, 11-day-old *hiz2* mutants grown on 0.5× MS medium displayed significantly shorter primary roots and nearly no lateral roots compared with WT (Fig. 6e–g). Under NPA and NAA treatment, *hiz2* mutants showed only a few very short induced lateral roots (Fig. 6h), reminiscent of that in *vir-1* plants (Fig. 3a). Trichome branching is known to be influenced by mRNA methylation[3,27,28]. The hypomorphic *mta* mutant has an increased

trichome branching phenotype[3], which is also observed in the *hiz2* mutants (Supplementary Fig. 11). The percentage of trichomes in *hiz2* mutants with more than three branches was 65% compared to 15% in WT, and 14% had five and more branches whereas WT had only 0.3% (Fig. 6i). Measurement of m⁶A levels in *hiz2* mutants revealed a decrease of 85% (Fig. 6j), which is comparable to the decrease of m⁶A levels in the major hypomorphic m⁶A writer mutants[3,8]. Thus, we conclude that HIZ2 is an important component of the *Arabidopsis* m⁶A writer complex. It is potentially a core component of the finely tuned mechanism of mRNA m⁶A modification involved in *Arabidopsis* growth and development.

**Fig. 6 Characterizing *Arabidopsis* HIZ2. a** Schematic of *HIZ2* (AT5G53440) genomic DNA sequence with T-DNA insertion sites. *HIZ2.1* and *HIZ2.2* refer to two splice variants. White rectangles denote UTRs, black rectangles denote exons and thick black lines represent introns. *hiz2-1* to *hiz2-3* refer to Salk_120590, Salk_020625 and Salk_126486, respectively. HIZ2qPCR-fwd-1 and HIZ2qPCR-rev-1 were used as the first primer pair while HIZ2qPCR-fwd-2 and HIZ2qPCR-rev-2 were the second primer pair in RT-qPCR. **b, c** *HIZ2* transcript levels checked by RT-qPCR using the first primer pair (**b**) and the second primer pair (**c**) labeled in **a**. *CBP20* was used as a reference gene. Data represent mean ± SE from three biological replicates. Statistically significant differences relative to WT here and in **f, g, j** were analyzed by One-Way ANOVA (one-sided test) and marked with asterisks (***$p < 0.001$). **d** Five-week-old *hiz2* mutants and WT planted in compost. Scale bar = 1 cm. **e** Eleven-day-old seedlings vertically cultured on 0.5× MS plus 1% sucrose. Scale bar = 1 cm. **f, g** Statistical data showing the root phenotypes of seedlings in **e**. Data represent mean ± SE ($n = 25$ individual seedlings). In box plots, the center line in each box indicates the median. The lower and upper bounds of each box represent the first quartile (25%) and the third quartile (75%), respectively. The bottom and top of whiskers denote the minimum and maximum, respectively. **h** Roots under the Stereo Dissecting Microscope after the treatment with *N*-1-naphthylphthalamic acid (NPA, 10 μM) and then transferred onto 0.5× MS with 1-naphthaleneacetic acid (NAA, 10 μM) for 5 days. Roots were soaked in 100% ethanol for a few seconds to minimize the effect of root hairs in photographing lateral roots. Scale bar = 500 μm. **i** Percentage of trichome branches on the first pair of rosette leaves from 2-week-old plants cultured in compost. Twenty leaves were used for counting the trichome branching in each line. **j** m$^6$A levels checked by two-dimensional thin layer chromatography (TLC) analysis. Data represent mean ± SE from three biological replicates. Source data are provided as a Source Data file.

## Discussion

Both the plant and the mammalian HAKAI are members of the m$^6$A writer complex[8,11]. The mammalian HAKAI was originally characterized as an E3 ubiquitin ligase[29], thus we thought that the plant homolog may carry out a similar function, and might regulate the m$^6$A writer complex members at a posttranslational level. The proteomics experiments revealed that HAKAI binds to two writer complex members, which we name HIZ1 and HIZ2. HAKAI's presence is essential for the HIZ1 binding to the core member MTA (METTL3), thus we suspected that the presence of HIZ1 in the complex may be regulated by posttranslational modification carried out by HAKAI. We did observe an increased amount of GFP-tagged HIZ1 in the absence of HAKAI but at the same time the transcript level of the *HIZ1pro:HIZ1-GFP* transgene was also increased, indicating HIZ1 is regulated by HAKAI, but not necessarily at a posttranslational level. Thus, whether the plant HAKAI possesses an E3 ligase function and if so, what the significance of this is, remains an open question.

To test whether the reduced level of m$^6$A seen in the *hakai* lines is due to a general reduction of methylation across all target transcripts, or results from loss of methylation from just a specific subset, we determined the topology of m$^6$A sites globally in WT, *hakai-2*, and two other complex member mutants, *fip37-4* and *vir-1*. The distribution of m$^6$A peaks was very similar between *hakai-2* and WT, and the number of peaks that were methylated in WT but which were absent or substantially reduced in *hakai-2* were fewer than those that were missing in the *fip37-4* and *vir-1* mutants. Thus, the majority of the 35% reduction in m$^6$A abundance seen in *hakai* mutants is likely due to a global reduction in methylation. However, a small number of transcripts appear to be disproportionally affected, and we found that those transcripts which are methylated in WT but not in *hakai-2* were enriched in association with root hair development processes. Although *hakai-2* does not show root hair phenotypic changes, the methylation changes in relation to root hair development may be significant, as the *HIZ1* overexpressor shows increased root hair length and density in response to NPA treatment in a HAKAI-dependent manner. When we compared m$^6$A topology between the three mutants, it was apparent that both *vir-1* and *fip37-4* lost most of the 3′ end m$^6$A peaks and had increase in 5′ end and middle peaks. As far as biological processes are concerned, transcripts that lost m$^6$A peaks in the 3′ UTRs were highly associated with embryo development and lateral root development, which was common between *fip37-4* and *vir-1*. One of the highest enrichment terms was mRNA polyadenylation and this was only found in the *fip37-4* dataset. An association between mRNA methylation and polyadenylation site choice was previously shown in the *vir-1* mutant line[24]. Parker et al. propose

that for certain transcripts, the proper sites for polyadenylation are directly labeled by a nearby m$^6$A[24]. We found that six transcripts that are involved in mRNA polyadenylation were missing m$^6$A peaks in the 3′ UTR (*PAPS2, PAPS4, CPSF160, CFIS2, CPSF73-I*) or the peaks were significantly reduced (*FY*) in the *fip37-4* mutants. Thus, we would like to suggest that the poly(A) synthesis and the cleavage site may also be altered by the m$^6$A dependent expression changes of the enzymes directly involved in this process.

Our study identified two *Arabidopsis* zinc finger proteins that co-purified with HAKAI, HIZ1 and HIZ2. Similar to the core complex member mutants, the knockout of HIZ2 results in decreased global m$^6$A levels, and very severe developmental defects. The mammalian CCCH-type zinc finger protein, ZC3H13, and its *Drosophila* homolog Female lethal 2 (Fl(2)d)-associated complex component (Flacc), have been identified as writer complex members[9,13]. The *Arabidopsis* HIZ2 is composed of a similar number of amino acid residues and possesses low-complexity regions, similar to Flacc (Supplementary Fig. 2). In addition, the lack of full CCCH domains in both Flacc and HIZ2 (Supplementary Fig. 2), suggests that HIZ2 is structurally more similar to the *Drosophila* Flacc than to the mammalian ZC3H13 protein. Disruption of Flacc function results in reduced m$^6$A level and defects in sex determination due to aberrant splicing of *Sex lethal (Sxl)*[9], similar to the knockout of the *Drosophila* METTL3 homolog Inducer of meiosis 4 (dIme4)[30,31]. The *Drosophila* Flacc is specifically involved in stabilizing the methylase complex, serving as a bridge between the RNA binding protein Spenito (Nito, homologous to *Arabidopsis* FPA) and Fl(2)d (FIP37 homolog)[9]. Depletion of mouse ZC3H13 impairs embryonic stem cell self-renewal and triggers differentiation[13]. In human, the functional role of ZC3H13 other than interacting with WTAP is poorly understood, although a recent report suggests that mutant ZC3H13 may facilitate glioblastoma progression[32,33]. However, whilst the *Drosophila* and the mammalian zinc finger homologs may seem to be involved in different processes and are structurally somewhat divergent, human ZC3H13 can restore the interaction between Nito and Fl(2)d upon deletion of endogenous *Drosophila* Flacc[9]. Although rather divergent in sequence and structure, the requirement for HIZ2, ZC3H13 and Flacc in the *Arabidopsis*, mouse and *Drosophila* methylation writer complex indicates a conserved role for a zinc finger protein. In *Arabidopsis*, HIZ2 is likely to be a functional homolog of *Drosophila* Flacc, and it interacts with MTA (METTL3) and HAKAI. However, the interaction of HIZ2 with either FIP37 or FPA is yet to be explored.

Another mammalian zinc finger protein which has been implicated in m$^6$A writing is mouse ZFP217, which belongs to the

$C_2H_2$-type zinc finger protein family, and acts as a multi-functional oncogene[16,34]. ZFP217 transcriptionally activates *Nanog* and *Sox2*, as well as other genes linked to the undifferentiated state of embryonic pluripotent stem cells[16]. In addition, through an interaction with the writer complex, ZFP217 inhibits $m^6A$ deposition in the same mRNA molecules. When ZFP217 is depleted, the expression of pluripotency factors decreases in parallel with a global increase of $m^6A$ methylation, which triggers the degradation of core stem cell transcripts[16]. Thus, ZFP217 regulates the balance between stem cell regeneration and differentiation. We found that the *Arabidopsis HIZ1* over-expression driven by the constitutive *35S* promoter led to a 24% reduction in global $m^6A$ abundance. Thus, we propose that the *Arabidopsis* HIZ1, via its binding to HAKAI, may be an inhibitor of mRNA methylation, similar to ZFP217. In addition, the GFP-tagged HIZ1 under its own promoter demonstrated the highest expression in stem cell niches in both primary and lateral root tips. Thus, the plant HIZ1 and the mouse ZFP217 show some degree of functional similarities in spite of their lack of a common structural or sequence relationship. However, it should be noted that we don't have evidences from knockout mutants to support the proposed function of HIZ1.

When *HIZ1* was overexpressed under the strong and constitutive *35S* promoter the plants showed some characteristics of the core writer complex mutants, for instance, decreased level of $m^6A$ and reduced lateral root induction in response to NPA and NAA treatment. When *HIZ1* was overexpressed under the control of its own promoter, the $m^6A$ inhibitory effect was not observed, neither was the decrease in lateral root induction. However, in the *HIZ1* overexpressor under *35S* promoter, and in plants transformed with *HIZ1* under its own promoter, we observed increased root hair density and lengths upon NPA treatment. This effect was strongest when the *HIZ1* native promoter was used, and under NPA treatment, this promoter drives strong GFP expression in root hairs. The increase in root hair length and density is likely acting through the key regulator RSL4, the expression of which increases in these plants. However, *RSL4* itself is not methylated in our data sets, thus, the mRNA methylation process probably indirectly regulates its expression under auxin efflux inhibition. Interestingly, a previous study using a panel of 166 geographically diverse *Arabidopsis* accessions to look at natural variation in root hair response to low phosphate conditions, identified a SNP which the authors attributed as causative of the root hair phenotype. This SNP resides between *HIZ1* and an upstream gene and they went on to show that root hair density (though not length) was altered in response to low phosphate in a *HIZ1* insertion mutant[35]. Neither root hair density nor length were altered in the mutant under phosphate replete conditions[35]. These results do however, indicate a natural function for HIZ1 in root hair formation, and as the root hair phosphate response is auxin-mediated[36], it may explain why the effects following NPA treatment are observed in the plants in which *HIZ1-GFP* is under the control of either the constitutive *35S* or its native promoter.

We propose a model where HIZ1 interacts with the methylation writer complex (via HAKAI) to fine tune methylation by down regulating methylase activity in those cells in which it is expressed, similar to the methylase suppressing activity of mouse ZFP217. In this scenario, strong and constitutive expression of HIZ1 would be expected to result in a global (and HAKAI dependent) reduction in $m^6A$, as we observe. However, it should be noted that the HIZ1 knockout lines do not show obvious root meristem defects that might be predicted to result if they contain higher $m^6A$ levels in the stem cell niche. An alternative possibility is that overexpression of HIZ1 disrupts the structure and stoichiometry of the writer complex. This seems less likely, as the HIZ1 knockout mutant has normal $m^6A$ levels (in whole

seedlings), and it is not strongly expressed outside of the meristematic and root hair cells, suggesting that it is not ubiquitously required as a key structural component.

Based on our findings, we propose that HAKAI acts as a "bridge", connecting HIZ1 with the core $m^6A$ writer components. In addition, HIZ2 is an essential member of the $m^6A$ writer complex required for full methylation of transcripts and is likely the plant equivalent of ZC3H13 (Flacc) of the mammalian (*Drosophila*) $m^6A$-METTL Associated Complex.

## Methods

**Plant materials and growth conditions.** *Arabidopsis* lines (ecotype Colombia-0) used include WT, $m^6A$ writer mutants and transgenic lines containing sequences encoding $m^6A$ writer proteins. Specifically, *mta ABI3:MTA*, *fip37-4* and *vir-1* are hypomorphic alleles of *MTA*, *FIP37* and *VIR*, respectively[8]. *hakai-2* is a HAKAI knockout line generated via CRISPR-Cas9 mutagenesis[8]. T-DNA insertion mutants, including Salk_045882 (*hiz1-1*), Salk_000717 (*hiz1-2*), Salk_120590 (*hiz2-1*), Salk_020625 (*hiz2-2*) and Salk_126486 (*hiz2-3*), were obtained from Nottingham Arabidopsis Stock Center (NASC) (Nottingham, UK).

*Arabidopsis* seeds were cultured in plates or compost depending on the subsequent use. Seeds planted in plates were sterilized by soaking in 5% NaClO for 4 min and then washed with sterile water for five times before being plated onto 0.5× MS medium plus 1% sucrose. Plates were placed in the cold room (4 °C, dark) for 2 days and then kept in the tissue culture room (16 h light: 8 h dark, 22 °C) until used. Plants grown directly in compost were maintained in the phytotron (16 h light: 8 h dark, 22 °C day: 18 °C night).

**Generation of transgenic lines.** Recombinant plasmid constructs generated in this study include *HAKAIpro:HAKAI-GFP*, *MTApro:MTA-GFP*, *FIP37pro:FIP37-GFP*, *HIZ1pro:HIZ1-GFP*, and *35Spro:HIZ1-GFP*. These constructs were generated by cloning the respective genomic DNA sequence (coding sequence for *MTA*) without the stop codon and 3′ UTR to Gateway entry vector pCR™8/GW/TOPO® (Invitrogen), then recombining into the appropriate pGreen-based GFP Gateway destination vector[37] using Gateway® LR Clonase™ II Enzyme Mix (Invitrogen). Transgenic lines were obtained by floral dip transformation[38] with the above recombinant constructs. Homozygous transgenic lines were selected and GFP-tagged protein localization in roots was determined via confocal microscopy (Leica TCS SP5). Primers used for cloning and screening were listed in Supplementary Data 9.

**Poly(A) RNA preparation and $m^6A$ measurement.** Total RNA was isolated using hot phenol extration[39]. Poly(A) RNA was prepared according to the NEBNext Poly(A) mRNA Magnetic Isolation Module (New England Biolabs, NEB). $m^6A$ levels were measured by two-dimensional thin layer chromatography (TLC) analysis as described in Zhong et al.[6]. Fifty nanogram of poly(A) RNA was digested by 1 µL of Ribonuclease T1 (1000 U µL⁻¹; Thermo Fisher Scientific) in 1× polynucleotide kinase (PNK) buffer A at 37 °C for 1 h. 5′ ends of digested RNA fragments were labeled with 0.5 µL of [γ-³²P]ATP (6000 Ci mmol⁻¹; PerkinElmer) using 10 U of T4 PNK. After overnight ethanol precipitation, the labeled RNA pellet was resuspended in 10 µL of 50 mM sodium acetate (pH 5.5) and digested by nuclease P1 (Sigma-Aldrich) at 37 °C for at least 1 h to produce mononucleotides. One microliter of the digested sample was loaded onto the cellulose F TLC plate (20 × 20 cm; Merck) and developed in a solvent system, with isobutyric acid:0.5 M $NH_4OH$ (5:3, v/v) as the first dimension buffer and isopropanol:HCl:water (70:15:15, v/v/v) as the second dimension buffer. The labeled nucleotides were identified and quantified by using a storage phosphor screen (Fuji-Screen) and Bio-Rad Molecular Imager FX system in combination with Quantity One software (v4.6.2.70).

**In vivo interaction proteomics-mass spectrometry analysis.** The in vivo interaction proteomics-mass spectrometry (IVI-MS) libraries were prepared according to the method described in Parker et al.[14]. Freshly harvested 2-week-old seedlings were crosslinked using 1% formaldehyde and the crosslinking was quenched by 0.125 M glycine solution. Nuclei were isolated from ground frozen sample powder using HONDA buffer (20 mM Hepes KOH pH 7.4, 10 mM $MgCl_2$, 440 mM sucrose, 1.25% Ficoll, 2.5% Dextran T40, 0.5% Triton X-100, 5 mM DTT, 5 mM phenylmethylsulfonyl fluoride [PMSF] and 1% Protease Inhibitor Cocktail [Sigma-Aldrich]). After centrifuging at 2000 × *g* for 17 min at 4 °C, the pellet was washed twice with HONDA buffer. The isolated nuclei samples were lysed using nuclei lysis buffer (50 mM Tris-HCl pH 8.0, 10 mM EDTA pH 8.0, 1% SDS, 1 mM PMSF and 1% Protease Inhibitor Cocktail [Sigma-Aldrich]) via sonication using a water bath sonicator (Diagenode Bioruptor 200) on low power with four cycles of 30 s ON and 60 s OFF. After centrifugation at 16,100 × *g* for 15 min at 4 °C, the supernatant was diluted using IP dilution buffer (16.7 mM Tris-HCl pH 8.0, 1.2 mM EDTA pH 8.0, 167 mM NaCl, 1.1% Triton X-100 and 1% Protease Inhibitor Cocktail [Sigma-Aldrich]) and then pre-washed GFP-Trap_A agarose beads (ChromoTek) was added to pull down target protein complex. The mixture was

incubated at 4 °C with rotation for 5 h. Afterwards, beads were washed three times using beads washing buffer (20 mM Tris-HCl pH 8.0, 150 mM NaCl, 2 mM EDTA pH 8.0, 1% Triton X-100, 0.1% SDS and 1 mM PMSF). Samples were incubated at 90 °C for 30 min to reverse formaldehyde crosslinking. After running samples on SDS-PAGE gel, the gel for the same lane was cut into five slices. Protein samples were digested into peptides by trypsin and then analyzed on LTQ Orbitrap Velos Pro mass spectrometer (Thermo Fisher Scientific). WT was used as a control and three biological replicates were performed for each line.

After mass spectrometry, raw files corresponding to five slices from the same lane were merged and analyzed via MaxQuant (v1.6.0.16) using label free quantification (LFQ)[40]. Unique and razor peptides were used for protein quantification and the protein and peptide false discovery rates (FDRs) were set to 0.01. MaxQuant output files containing proteins that were identified in each sample were subsequently analyzed using Perseus (v1.6.15.0)[41]. Potential interacting proteins were determined using two-sided student $t$-test and visualized by volcano plots. The significance was analyzed based on the negative logarithm of the $p$-value derived from the $t$-test ($-\log_{10}p$-value). Threshold values (FDR) were set between 0.01–0.1 and S0 (curve bend) at 1 or 2. In volcano plots, significantly-enriched proteins were separated from others by a hyperbolic curve.

**Gene expression analysis by RT-qPCR.** Total RNAs were extracted from 2-week-old seedlings (for confirming T-DNA inserted mutations and the expression levels of transgenes) or from root samples under different treatments. Total RNAs were treated with Ambion® TURBO™ DNase (Invitrogen) and then purified via GeneJET RNA Cleanup and Concentration Micro Kit (Thermo Fisher Scientific). First-strand cDNAs were synthesized using ProtoScript II Reverse Transcriptase (NEB). qPCR was carried out on LightCycler® 480 instrument (Roche) using SensiMix™ SYBR® Low-ROX (Bioline). Three biological replicates were analyzed for each sample. Relative expression levels were determined using the $2^{-\Delta\Delta CT}$ method[42]. *Arabidopsis CBP20* (AT5G44200) and *UBC21* (AT5G25760) were used as internal controls. qPCR primer pairs were listed in Supplementary Data 9.

**Western blot.** Proteins were extracted from 2-week-old seedlings cultured on 0.5× MS medium using the nuclei lysis buffer as mentioned above. Protein samples were separated on Bio-Rad Mini-PROTEAN® TGX™ Precast Gel and transferred onto Amersham™ Protran® western blotting membrane (nitrocellulose, pore size: 0.2 μm). Subsequent steps were performed following instructions for 'WesternBreeze® Chemiluminescent Western Blot Immunodetection' Kit (Invitrogen). The GFP antibody used is a mixture of two monoclonal antibodies from mouse IgG1κ (clone 7.1 and 13.1, Sigma-Aldrich) at a 1:1000 dilution. An anti-actin antibody, a rabbit polyclonal antibody targeting a set of actins in *Arabidopsis* (Agrisera) was used as a control (at 1:2500 dilution). The protein amounts were quantified using Quantity One software (v4.6.2.70). Uncropped scans for western blot membranes were included in the Source Data file or Supplementary Information file.

**Analysis of root phenotypes.** *Arabidopsis* primary root lengths and the number of lateral roots were recorded from 10-day-old seedlings vertically cultured on 0.5× MS with 1% sucrose. Primary root lengths were measured using ImageJ software (v1.8.0_172). Pericycle activation for lateral root initiation was performed according to the method described in Himanen et al.[21]. Specifically, 4-day-old *Arabidopsis* seedlings vertically cultured on 0.5× MS were transferred to vertical square plates containing 0.5× MS and 10 μM NPA. On the 3rd day of culturing on NPA, they were transferred onto 0.5× MS plus 10 μM NAA. To observe root hair morphology on NPA, seedlings grown on NPA remained growing on the same medium after the 3rd day and images of root hairs were taken on the 6th day of culturing on NPA. Roots were photographed under the Stereo Dissecting Microscope (Zeiss Stemi SV6). The number and lengths of root hairs were measured using ImageJ software (v1.8.0_172). Statistical analysis was performed using GraphPad Prism (v9.2.0).

**GUS expression assay.** Harvested roots were immersed in GUS staining reagent mix (50 mM phosphate buffer, 10 mM EDTA, 20% methanol, 0.1% TritonX-100 and 1 mg mL⁻¹ X-Gluc) while shaking for 12 h. Afterwards, stained roots were soaked in 100% ethanol for destaining. Photos were taken under the Stereo Dissecting Microscope (Zeiss Stemi SV6).

**MeRIP-Seq and differential expression analysis.** Total RNA from three replicates of 2-week-old seedling grown on 0.5× MS was isolated using hot phenol extraction[39]. This was followed by one-round poly(A) purification using oligo d(T) magnetic beads (NEB). 1.5–2 μg of mRNA was fragmented to 100–150 nts using RNA Fragmentation Reagent (Thermo Fisher Scientific) followed by overnight ethanol precipitation. After centrifugation and washing, the pellets were resuspended in 10 μL of H₂O. Nine microliter of the solution was used for immunoprecipitation, and one microliter for preparing the input libraries. The fragmented RNA was mixed with Protein G magnetic bead prebound monoclonal anti-m⁶A antibody (1 μL) from the EpiMark N6-Methyladenosine Enrichment Kit (NEB), resuspended in 300 μL EpiMark IP buffer supplemented with murine RNase

inhibitor (NEB). All following steps were as described by the manufacturer. After the last wash we carried out an extra washing step using H₂O. We omitted the final elution step as we carried out the cDNA synthesis on the magnetic beads using ScriptSeq v2 RNA-Seq Library Preparation Kit (Illumina). The libraries for both IP and input were size selected using E-Gel SizeSelect II (Invitrogen), and quality checked on Agilent High Sensitivity DNA Chips (Agilent). The pooled libraries were sequenced on Nextseq 500 (Illumina) at the Nottingham University's Sequencing Service.

For the MeRIP analysis, sequencing reads contaminating adapter sequences and low quality reads (phred scores <30) were removed using Trimgalore (v0.4.4, https://www.bioinformatics.babraham.ac.uk/projects/trim_galore/). The processed fastq reads were aligned to the Ensembl annotated TAIR10.49 reference genome using STAR (v2.5.0)[43]. The resultant bam files were indexed using SAMtools (v1.10, PMID: 19505943)[44] and m⁶A enriched regions identified in m⁶A immunoprecipitated samples over inputs using m6aViewer (v1.6.1) (PMID: 28724534)[45]. BEDtools (v2.27.1, PMID: 20110278)[46] was used to extend peaks by 100 bp upstream and downstream. Only those peaks represented in at least two replicates, and 4-fold enriched were taken forward for further analysis using the RNAmod, interactive web-based platform (http://61.147.117.195/RNAmod/) with default settings[47]. Using the Peak matrix dataset created by the RNAmod platform, we produced lists of peaks which were present in WT samples and missing from the mutant data. The GO enrichment analysis on these gene lists was carried out using PANTHER (v16.) Classification System website (http://pantherdb.org).

For differential expression the MeRIP-Seq input files qualities were assessed by fastqc (http://www.bioinformatics.babraham.ac.uk/projects/fastqc) prior to trimming with Trimmomatic (v0.38)[48]. Trimmed reads were aligned to the TAIR10 *Arabidopsis* genome (TAIR10.48) using Bowtie2 (v2.4.2)[49] with local alignment enabled. Aligned bam files were indexed with SAMtools (v1.10)[44] prior to gene counting with HTSeq (v0.9.1)[50]. A matrix of gene counts was generated from the read counts and imported into R-studio (v1.4.1717-3, https://www.rstudio.com/). Differential expression analysis was performed using DESeq2 (v2.11.40.6)[51]. To generate a list of differentially expressed genes, we chose genes with a log2 fold change above 1.5 (enriched) or below −1.5 (depleted) and adjusted $p$-value of less than 0.05. The default tests for calculating $p$-value (Wald) and correcting for multiple testing (Benjamini and Hochberg) were used. We identified gene list overlaps between our methylation datasets and differential expression data from our input samples as well as from published expression data[25,26]. To calculate the statistical significance, the online tool (http://nemates.org/MA/progs/overlap_stats.html) was used.

**Analysis of trichome branching.** The number of trichome branches were counted from the first pair of rosette leaves of 2-week-old *Arabidopsis* plants cultured in compost. Trichomes with different number of branches were photographed under the Stereo Dissecting Microscope (Zeiss Stemi SV6).

**Reporting summary.** Further information on research design is available in the Nature Research Reporting Summary linked to this article.

## Data availability

Sequence details for *Arabidopsis* genes used in this study can be found in TAIR database (https://www.arabidopsis.org/) with the following accession numbers: AT5G01160 (*HAKAI*), AT1G32360 (*HIZ1*), AT5G53440 (*HIZ2*), AT4G10760 (*MTA*), AT3G54170 (*FIP37*) and AT3G05680 (*VIR*). Protein sequences for possible HIZ2 homologs in other organisms used in this study can be found in the UniProt Knowledgebase (UniProtKB, https://www.uniprot.org/) with the following identifiers: Q5T200-1 (Human ZC3H13), E9Q784-1 (Mouse ZC3H13), and Q9VWN4-1 (*Drosophila* Flacc). The mass spectrometry proteomics data generated in this study have been deposited in the ProteomeXchange Consortium via the PRIDE[52] partner repository under the dataset identifier PXD026287. The MeRIP-Seq data generated in this study have been deposited at NCBI's Gene Expression Omnibus (GEO) under the accession number GSE174573. The transcriptomics data generated in this study have been deposited at GEO under the accession number GSE188610. Source data are provided with this paper.

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

## Acknowledgements

This work was supported by Biotechnology and Biological Sciences Research Council (BBSRC) grants (grants BB/M008606/1 and BB/S006478/1) awarded to R.G.F. and by BBSRC grant (BB/M010066/1) awarded to G.G.S. The PhD program of M.Z. was supported by China Scholarship Council Research Excellence Scholarship and the University of Nottingham Future Food Beacon Doctoral/Post-Doctoral Prize.

## Author contributions

M.Z., Z.B., K.M., and S.Z. conducted experiments, Z.B., N.P.M., and N.A. carried out bioinformatics analyses. R.G.F. conceived research and designed experiments. M.Z., Z.B., and R.G.F. wrote the manuscript. G.G.S. and N.P.M. edited manuscript. All authors read and commented on the final version of the manuscript.

## Competing interests

The authors declare no competing interests.
