## [Peer Review File · Nature Communications]

Two zinc finger proteins with functions in m6A writing interact with HAKAIReviewers' Comments:

Reviewer #1:

Remarks to the Author:

This is an interesting and timely study that uses proteomic analyses to identify new components of the methylation complex in Arabidopsis. They do this by finding proteins that interact with the complex component HAKAI. The proteins they name HIZ1 and HIZ2 and they seem to have opposite effects. With HIZ1 appearing to inhibit methylation and HIZ2 appearing to be a core component of the plant methylation machinery and thus necessary for proper RNA methylation in Arabidopsis. The plants containing mutations in both of these newly identified proteins display interesting phenotypes and some follow up analyses of these mutant plants are provided, but not to the level that is satisfying for me in this study. I thus outline my necessary and suggested experimental revisions for this study below.

NECESSARY EXPERIMENTAL REVISIONS

The main thing this manuscript lacks and needs added to a revised manuscript is a connection between the phenotypes of the hiz mutants and over-expresser lines and the methylated transcripts that are likely to affect the phenotypes displayed by these mutant plants. Some connection between transcripts of known lateral root and/or root hair/trichome phenotypes and changes in methylation levels and transcript abundance/stability would be an important addition to this manuscript. The identification of these new methylation components is a nice advance but directly.

SUGGESTED EXPERIMENTAL REVISIONS

Experiments providing stronger support for a negative effect on methylation of HIZ1 and demonstrating the opposite functionality for HIZ2 would be a nice addition to this manuscript. If the group chooses to do these experiments based on other Reviewer comments, then I would recommend saving an in depth phenotypic and plant developmental biology study of the hiz1 and hiz2 mutant plants for a later manuscript where this study can directly link these phenotypes to specific methylated transcripts (as I outlined in the section above).

Reviewer #2:

Remarks to the Author:

In the manuscript, the authors reported that two zinc finger proteins, HIZ1 and HIZ2, interacted with HAKAI, a member of the m6A writer complex, but functioned in different manners. m6A levels were not altered in hiz1 mutants, but were reduced in plants overexpressing HIZ1. While HIZ1 was isolated as an interacting partner of HAKAI1, HAKAI1 possibly affected the mRNA expression of HIZ1. Moreover, the authors showed that reducing m6A in various genetic materials pertaining to m6A modification might be associated with altered auxin responses in lateral root development. They further performed MeRIP-seq using WT, hakai-2, fip37-4 and vir-1 mutants and compared the transcriptome-wide m6A topology in these genotypes. In addition, the authors also found an 85% decrease in m6A levels in hiz2 mutants, which show strong developmental defects similar to what observed in other known hypomorphic m6A writer mutants.

Although this manuscript includes many pieces of information regarding m6A writers, the whole manuscript lacks focal points to be addressed. Each piece of information is interesting, but all of them lack in-depth mechanisms that justify their publications in a good journal. Many results are premature, and need to be further characterized. My major concerns are summarized below:

1. The molecular functions of HIZ1 and HIZ2 remain largely unclear based on the results provided in this study. Characterization of HIZ1 is mostly based on its overexpression phenotype, which questions the actual function of HIZ1 in vivo. Furthermore, there is no change in m6A levels in HIZ1 knock out mutants. These results do not support that HIZ1 is required for m6A deposition under normal

conditions.

2. Knocking out HIZ2 shows interesting phenotypes and altered m6A levels. But this mutant is not characterized further. What are the targets of HIZ2? How does it interact with other writers to affect m6A?

3. The authors show that reducing m6A in several m6A writer mutants affects auxin responses and root development. While the physiological and phenotypic analyses in this part are interesting, the concrete mechanisms that link m6A and auxin response are not clear.

4. The transcriptome-wide analysis of m6A topology in WT, *hakai-2*, *fip37-4* and *vir-1* does not provide new insights into the effects of m6A in plants and the function of individual writer components compared to what has been known in the field. More detailed bioinformatics analysis might be required.

5. For the conclusion that HAKAI mediates the interaction between HIZ1 and MTA, more pieces of evidence, including pull-down and/or CoIP assays and genetic experiments, should be included.

Reviewer #3:

Remarks to the Author:

Zhang and coauthors have presented a thorough analysis of two novel zinc finger proteins, HIZ1 and HIZ2, which have essential roles in the m6A writer complex and root development in Arabidopsis. The use of genetic phenotyping combined with fluorescence microscopy, immunoprecipitation, mass spectrometry, and other methods produce complementary evidence that paints a compelling picture in support of the authors conclusions that HIZ2 is a critical member of the Arabidopsis m6A writer complex while HIZ1 is a HAKAI-dependent negative regulator of m6A deposition. This work is a significant step forward toward the understanding of mRNA methylation in Arabidopsis and will be well received by the community of associated plant scientists.

The following suggestions will help further improve the manuscript:

Major:

- The authors elected to use a 10% FDR cutoff for the statistical significant thresholds in their IVI-MS experiments. This is quite high; most papers use a 1% or 5% threshold. Can the authors clarify their rationale for using a 10% cutoff? Are their target proteins still identified when they apply a more rigorous FDR threshold?
- The authors have not included any of their methods for LC-MS/MS analysis or parameters for database searching. Along these lines, it is difficult to understand if the authors used a bottom-up or top-down proteomics approach. This should be amended before publication.
- The authors should clearly define their identification criteria for significantly enriched proteins in IVI-MS experiments. Several identified proteins in their supplemental table contain 0 unique peptides detected. How do the authors rationalize the identify of these proteins?
- Within HIZ1 knockout plants, the authors report normal levels of m6A compared to wild type plant lines. If HIZ1 is a negative regulator of m6A deposition, wouldn't there be an expectation of higher levels of m6A in these knockout plants? Can the authors provide additional rationale for this result?
- Have the authors considered co-immunoprecipitation experiments with HIZ1/HIZ2 to directly analyze their interacting proteins?
- Have the authors considered performing similar experiments to what is explained on page 12, lines 331-334 (complementing *Drosophila* Flacc knockouts with ZC3H13) to provide additional evidence that HIZ2 is a functional homolog of *Drosophila* Flacc?
- It is hard to follow the rationale behind increased root hair density and length in plants transformed with HIZ1 under its own promotor. Why is RSL4 expression increased in these plants? The authors mention that mRNA methylation could indirectly regulate RSL4 expression, but m6A levels between

wild type, HIZ1pro:HIZ1-GFP/hiz1, and HIZ1pro:HIZ1-GFP/WT plant lines were similar. It is also difficult to understand the connection between the identified SNP upstream of HIZ1 under low phosphate conditions (page 13, lines 363-365) and the results in plants following NPA treatment where HIZ1-GFP is under the control of its native promoter. Could the authors please clarify these points within the text?

- The authors should include the full, unaltered western blot images in their supplementary materials. Were these conducted in replicates? What steps were taken to ensure actin was a true housekeeping gene in the mutant strains?

Minor:

- It seems that in IVI-MS experiments proteins were crosslinked *in vivo* based on provided reference 14. The authors should describe this in the manuscript to explain how they ensured that they identified *in vivo* interactions rather than interactions that may have occurred throughout the extraction and preparation processes.

- Reference 14 should be replaced with its published version (Elife).

- The authors should consider adding the protocol for IVI-MS and nuclear fractionation, regardless of publications elsewhere.

- On page 8, line 213: it would be helpful for readers to include the meaning of peaks when conducting MeRIP-Seq, as this is not a particularly common technique across disciplines.

- On page 9, it is unnecessary to include the p-values for each GO term. If included, it should be listed for all terms presented rather than selected terms.

- The fluorescence in extended data figure 1b appears to be oversaturated. Are the authors sure these images were measured within the analytical range of the microscope?

- The authors should provide full names for all protein acronyms (e.g., HAKAI, MTA) when first mentioning those proteins.

- The authors pose the scenario that the reduced level of m6A seen in the hakai lines could be due to a general reduction of methylation across all target transcripts, or from loss of methylation from certain proteins. While the authors provide their MeRIP-Seq results with GO enrichment analysis in this section, they should be more explicit with their conclusion of the reason behind this reduced level of m6A.

- On page 11, lines 296-297: could the authors please clarify what they mean by "...and the number of peaks that were methylated in WT and disappeared or were substantially reduced in hakai-2 were the lowest compared to the other two mutants" in the text?

Reviewer #4:

Remarks to the Author:

This is a well done work from Fray and colleagues who have been working on m6A RNA methylation in plants for decades. In this manuscript they identified two new components of the plant methyltransferase complex, HIZ1 and HIZ2, both interact with HAKAI. Similar to roles of known zing finger proteins in animals HIZ1 appears to suppress m6A methylation whereas HIZ2 is quite important to RNA m6A methylation. The authors perform studies of mutants and overexpression systems, respectively. I thought this is a well done work. I have a few suggestions:

For HIZ1 I am curious about its expression during plant development. I wonder if the authors can check published RNA-seq data from different stages of development. I am curious what type of transcripts it tend to suppress methylation and functional implications of this suppression. I thought data mining and analysis can be quite useful for the community.

I would suggest the same for HIZ2. I wonder if the authors have at least RNA-seq data to compare HIZ2 mutant versus wt. A comparison could provide valuable information.

A minor suggestion, in the introduction I would change fine tuning in "N6-methyladenosine (m6A) is

necessary for the fine tuning of developmental and cell differentiation processes..." to regulating. M6A is quite important to mammalian development.

Answers to the Reviewers' comments and suggestions

We would like to begin by thanking the reviewers for their comments and suggestions on how to improve our manuscript. All the changes introduced have been highlighted in the revised submission. A point-by-point response to the Reviewers' comments is presented below.

Reviewer 1

The main thing this manuscript lacks and needs added to a revised manuscript is a connection between the phenotypes of the *hiz* mutants and over-expresser lines and the methylated transcripts that are likely to affect the phenotypes displayed by these mutant plants. Some connection between transcripts of known lateral root and/or root hair/trichome phenotypes and changes in methylation levels and transcript abundance/stability would be an important addition to this manuscript. The identification of these new methylation components is a nice advance but directly.

We have carried out additional transcriptomic analysis as suggested and have added the following text:

We also wanted to see if there was any connection between the methylation levels of any transcript and their abundance. We carried out a transcriptomic analysis of all three mutants where we identified differentially expressed genes (1.5-fold change cut off). We then looked at the proportional representation of the transcripts that lost their methylation in these gene lists. We did not find any over representation of lack of methylation in the up or down regulated sets in any of the mutants (as an example, *vir-1* analysis is in Supplementary Data 4). Next, we used the dataset from Lavenus et al.²⁵ where the authors specifically identified genes that are regulated by auxin response factor 7 (ARF7), a transcription factor which is required together with ARF19 for lateral root initiation in response to auxin. We extracted all of the methylated genes from this dataset, which gave us a list of 90 m⁶A methylated transcripts, more than half (52%) of the verified ARF7 responsive gene dataset (173 genes) (Supplementary Data 5). We again wanted to know whether the loss of methylation in ARF7 responsive transcripts would affect their expression at the transcriptional level. Therefore, we extracted those genes that lose methylation in the *vir-1* mutant (a list of 19 genes, Supplementary Data 5). Out of the 19 transcripts that lost methylation, only two were found in the *vir-1* differentially expressed gene list. Nevertheless, these two transcripts *ARF19* and

auxin resistant 1 (*AUX1*) are both instrumental in lateral root development and have decreased expression in *vir-1*. This finding is in agreement with defective lateral root development phenotypes we observe in m⁶A deficient plants⁸. Next we looked at how methylation affects gene expression during root hair development. From a gene expression dataset of Lan et al.²⁶ that reports a root hair dataset compared with data obtained from analyzing all root tissues, we identified 365 methylated transcripts out of 635 up-regulated genes. This equates to a 57% representation (representation 1, p<0.152) of the mRNA methylation in those transcript that increase abundance during root hair development (Supplementary Data 6). Next we looked at the representation of methylated transcripts in the down regulated list of genes, which was 72.5%, a statistically significant increase (representation 1.3, p<1.042e-29, http://nemates.org/MA/progs/overlap_stats.html) (Supplementary Data 6). This suggests that during root hair development, the presence of m⁶A is associated with a decrease in abundance of the relevant transcripts.

SUGGESTED EXPERIMENTAL REVISIONS

Experiments providing stronger support for a negative effect on methylation of HIZ1 and demonstrating the opposite functionality for HIZ2 would be a nice addition to this manuscript. If the group chooses to do these experiments based on other Reviewer comments, then I would recommend saving an in depth phenotypic and plant developmental biology study of the *hiz1* and *hiz2* mutant plants for a later manuscript where this study can directly link these phenotypes to specific methylated transcripts (as I outlined in the section above).

m⁶A measurements on additional independent *35Spro:HIZ1-GFP* lines have been included in Fig 2c to highlight the consistent decrease in m⁶A observed.

As the Reviewer suggests, we plan a detailed phenotypic and plant developmental biology study for a later manuscript.

Reviewer 2

1. The molecular functions of HIZ1 and HIZ2 remain largely unclear based on the results provided in this study. Characterization of HIZ1 is mostly based on its overexpression phenotype, which questions the actual function of HIZ1 in vivo. Furthermore, there is no change in m⁶A levels in HIZ1 knock out mutants. These results do not support that HIZ1 is required for m⁶A deposition under normal conditions.

As we show in Fig 2d and Supplementary Fig 4, HIZ1 expression is restricted to a small number of cells (root hair cells, trichoblast cells, and meristematic cells of the root tip). These cells will only contribute a small proportion of the RNA to any whole seedling extract, and

provide insufficient RNA for harvest and assay in their own right. Thus we would not expect to be able to detect a small (~25%) increase in m⁶A levels that might be predicted to occur in these cells in the knockout mutant line.

The molecular function of HIZ2 within the complex remains unclear, but this is also true of the presumed human counterpart ZC3H13. However, the finding that its presence in the complex is conserved between humans, *Drosophila* and plants, even though the sequence conservation is rather poor compared to other complex members, is an important finding that will be of broad interest to the field.

2. Knocking out HIZ2 shows interesting phenotypes and altered m⁶A levels. But this mutant is not characterized further. What are the targets of HIZ2? How does it interact with other writers to affect m⁶A?

See comments above

3. The authors show that reducing m⁶A in several m⁶A writer mutants affects auxin responses and root development. While the physiological and phenotypic analyses in this part are interesting, the concrete mechanisms that link m⁶A and auxin response are not clear.

See response to Reviewer 1

4. The transcriptome-wide analysis of m⁶A topology in WT, *hakai-2*, *fip37-4* and *vir-1* does not provide new insights into the effects of m⁶A in plants and the function of individual writer components compared to what has been known in the field. More detailed bioinformatics analysis might be required.

The intention of this manuscript was to examine the role of HAKAI in more detail, and to test whether only a subset of transcripts are affected in the knockout mutant. Some transcriptomic analysis has been previously undertaken on hypomorphic *fip37* and *vir* mutants, but in order to allow comparison to our *hakai* mutant, we carried out our own MeRIP experiments of all WT and mutant lines grown under identical conditions.

We have carried out some additional bioinformatics analysis (see response to Reviewer 1).

5. For the conclusion that HAKAI mediates the interaction between HIZ1 and MTA, more pieces of evidence, including pull-down and/or CoIP assays and genetic experiments, should be included.

The experiments described in the results section are CoIP experiments. Both MTA and HAKAI bring down HIZ1. Furthermore, HIZ1 is no longer brought down by MTA in the absence of HAKAI. In addition, the genetic experiments in which HIZ1 OE phenotypes are lost upon crossing to the *hakai* mutant background further support this as a functional interaction.

Reviewer 3

The following suggestions will help further improve the manuscript:

Major:

- The authors elected to use a 10% FDR cutoff for the statistical significant thresholds in their IVI-MS experiments. This is quite high; most papers use a 1% or 5% threshold. Can the authors clarify their rationale for using a 10% cutoff? Are their target proteins still identified when they apply a more rigorous FDR threshold?

We have reanalyzed the data and regenerated significantly pulled-down protein list by increasing the FDR threshold to 5% and 1% (Fig.1f-h; Supplementary Data 1). Our target proteins remain in the pull-down list at this more stringent FDR cutoff. For identifying HAKAI interacting proteins, the FDR cutoff remained set at FDR<10% to keep the known interactor MTA among the pulled-down proteins with HAKAI (Fig.1e). In addition, these figures have been redrawn using the updated Perseus (version 1.6.15.0) software.

- The authors have not included any of their methods for LC-MS/MS analysis or parameters for database searching. Along these lines, it is difficult to understand if the authors used a bottom-up or top-down proteomics approach. This should be amended before publication.

Added in Methods and detailed parameters set for MaxQuant are included in Source Data – Fig. 1e-h Parameters.

New text; Freshly harvested 2-week old seedlings were crosslinked using 1% formaldehyde and the crosslinking was quenched by 0.125 M glycine solution. Nuclei were isolated from ground frozen sample powder using HONDA buffer (20 mM Hepes KOH pH 7.4, 10 mM MgCl₂, 440 mM sucrose, 1.25% Ficoll, 2.5% Dextran T40, 0.5% Triton X-100, 5 mM DTT, 5 mM PMSF and 1% Protease Inhibitor Cocktail). After centrifuging at 2,000 g for 17 min at 4°C, the pellet was washed twice with HONDA buffer. The isolated nuclei samples were lysed using nuclei lysis buffer (50 mM Tris-HCl pH 8.0, 10 mM EDTA pH 8.0, 1% SDS, 1 mM PMSF and 1% Protease Inhibitor Cocktail) via sonication using a water bath sonicator (Diagenode Bioruptor 200) on low power with 4 cycles of 30 s ON and 60 s OFF. After centrifugation at 16,100 g for 15 min at 4°C, the supernatant was diluted using IP dilution buffer (16.7 mM Tris-HCl pH 8.0, 1.2 mM EDTA pH 8.0, 167 mM NaCl, 1.1% Triton X-100 and 1% Protease Inhibitor Cocktail) and then pre-washed GFP-Trap_A agarose beads (ChromoTek) was added to pull down target protein complex. The mixture was incubated at 4°C with rotation for 5 h. Afterwards, beads were washed 3 times using beads washing buffer (20 mM Tris-HCl pH 8.0, 150 mM NaCl, 2 mM EDTA pH 8.0, 1% Triton X-100, 0.1% SDS and 1 mM PMSF). Samples were incubated at 90°C for 30 min to reverse formaldehyde crosslinking. After running samples on SDS-PAGE gel, the gel for the same lane was cut into 5 slices. Protein samples were digested into peptides by trypsin and then analyzed on LTQ

Orbitrap Velos Pro mass spectrometer (Thermo Fisher Scientific). WT was used as a control and 3 biological replicates were performed for each line.

After mass spectrometry, raw files corresponding to five slices from the same lane were merged and analyzed via MaxQuant (version 1.6.0.16) using label free quantification (LFQ)⁴⁰. Unique and razor peptides were used for protein quantification and the protein and peptide false discovery rates (FDRs) were set to 0.01. MaxQuant output files containing proteins that were identified in each sample were subsequently analyzed using Perseus (version 1.6.15.0)⁴¹. Potential interacting proteins were determined using student t-test and visualized by volcano plots. The significance was analyzed based on the negative logarithm of the p-value derived from the t-test ($-\log_{10}^{p\text{-value}}$). Threshold values (FDR) were set between 0.01-0.1 and S0 (curve bend) at 1 or 2. In volcano plots, significantly-enriched proteins were separated from others by a hyperbolic curve.

- The authors should clearly define their identification criteria for significantly enriched proteins in IVI-MS experiments. Several identified proteins in their supplemental table contain 0 unique peptides detected. How do the authors rationalize the identify of these proteins?

Added in Methods and detailed parameters set for MaxQuant are included in Source Data – Fig. 1e-h Parameters. We selected “unique and razor peptides” for protein quantification in Mass spec data analysis via Maxquant. This is why several identified proteins contain 0 unique peptides but they still have razor peptides. (Included in new text above).

- Within HIZ1 knockout plants, the authors report normal levels of m6A compared to wild type plant lines. If HIZ1 is a negative regulator of m6A deposition, wouldn't there be an expectation of higher levels of m6A in these knockout plants? Can the authors provide additional rationale for this result?

See response to Reviewer 2 point 1 first paragraph.

- Have the authors considered co-immunoprecipitation experiments with HIZ1/HIZ2 to directly analyze their interacting proteins?

- Have the authors considered performing similar experiments to what is explained on page 12, lines 331-334 (complementing *Drosophila Flacc* knockouts with ZC3H13) to provide additional evidence that HIZ2 is a functional homolog of *Drosophila Flacc*?

These experiments may form the base of a future study. The possibility of *Drosophila Flacc* complementing *hiz2* would be exciting, but these experiments and subsequent crosses would require many months to complete with a very uncertain outcome. We have tested the ability of other components of the human methylase complex to complement their plant counterparts, but these were not successful.

- It is hard to follow the rationale behind increased root hair density and length in plants transformed with HIZ1 under its own promoter. Why is RSL4 expression increased in these plants? The authors mention that mRNA methylation could indirectly regulate RSL4 expression, but m6A levels between wild type, HIZ1pro:HIZ1-GFP/hiz1, and HIZ1pro:HIZ1-GFP/WT plant lines were similar. It is also difficult to understand the connection between the identified SNP upstream of HIZ1 under low phosphate conditions (page 13, lines 363-365) and the results in plants following NPA treatment where HIZ1-GFP is under the control of its native promoter. Could the authors please clarify these points within the text?

HIZ1 under its own promoter is expressed in trichoblast and root hair cells (Supplementary Fig 4) (as well as the root tip). Added text below;

HIZ1pro:HIZ1-GFP is expressed in hair cells and this expression is strongly elevated following NPA treatment. In contrast, *35Spro:HIZ1-GFP* is expressed in all root cells (including root hairs) (Supplementary Fig. 4). Consistent with the root hair phenotypes, *HIZ1-GFP* transgene abundance in *HIZ1pro:HIZ1-GFP/hiz1* increased by 2 fold relative to the WT *HIZ1* transcript level (Supplementary Fig. 7). This suggests that the *HIZ1-GFP* transgene not only complements the loss of *HIZ1* expression in *hiz1-1* but also leads to the local overexpression of *HIZ1* to some extent, possibly due to the lack of regulatory sequences from the endogenous 3' UTR (Supplementary Fig. 3a), which is itself methylated and is absent in the carboxy-terminal GFP tagged lines.

RSL4 is a marker for root hair differentiation. Its induction is indicative that root hair specific pathways have indeed been induced;

The induction of *RSL4* is indicative of the initiation of root hair specific pathways in the NPA-treated *HIZ1-GFP* lines. However, this root hair promoting activity of HIZ1 appears to require the presence of HAKAI, as the positive effect on root hair development was lost when the *HIZ1-GFP* lines were crossed into genetic backgrounds in which HAKAI was knocked out (Supplementary Fig. 8).

The previously described SNP upstream of *hiz1* is expanded upon in the sentence below;

Interestingly, a previous study using a panel of 166 geographically diverse *Arabidopsis* accessions to look at natural variation in root hair response to low phosphate conditions, identified a SNP which the authors attributed as causative of the root hair phenotype. This SNP resides between *HIZ1* and an upstream gene

- The authors should include the full, unaltered western blot images in their supplementary materials. Were these conducted in replicates? What steps were taken to ensure actin was a true housekeeping gene in the mutant strains?

Original uncropped western blot images have been included in the Source Data for Fig. 2f, Supplementary Fig. 1c and Supplementary Fig. 3c. Three replicates were done for those showing differences (Fig. 2f and Supplementary Fig. 3c). Our western blot results and GFP-tagged protein expression checked by confocal microscopy mutually support each other. The anti-actin antibody we used is one suitable for plants, which is buffered against effects of one actin changing. Its reactivity has been confirmed in *Arabidopsis*. Validation of this antibody and previous publications using this antibody can be found in the manufacturers' websites (<https://www.agrisera.com/en/artiklar/act-actin.html>). The detail of this actin is as follows.

Actin - ca. 100 amino acids of recombinant actin conserved more than 80 % in *Arabidopsis thaliana*: actin-1 P0CJ46 AT2G37620, actin-2 Q96292 AT3G18780, actin-3 P0CJ47 AT3G53750, actin-4 P53494 AT5G59370, actin-5 Q8RYC2 At2g42100, actin-7 P53492 At5g09810, actin-8 Q96293 AT1G49240 , actin-11 P53496 , AT3G12110 ,actin-12 P53497 AT3G46520

Minor:

- It seems that in IVI-MS experiments proteins were crosslinked in vivo based on provided reference 14. The authors should describe this in the manuscript to explain how they ensured that they identified in vivo interactions rather than interactions that may have occurred throughout the extraction and preparation processes.

- Reference 14 should be replaced with the its published version (Elife).

- The authors should consider adding the protocol for IVI-MS and nuclear fractionation, regardless of publications elsewhere.

The updated reference has been included and the IVI-MS protocol included.

- On page 8, line 213: it would be helpful for readers to include the meaning of peaks when conducting MeRIP-Seq, as this is not a particularly common technique across disciplines.

We have changed text to MeRIP-Seq enriches for m⁶A-containing mRNA fragments, which are defined as peaks relative to the input sample.

- On page 9, it is unnecessary to include the p-values for each GO term. If included, it should be listed for all terms presented rather than selected terms.

Removed as suggested

- The fluorescence in extended data figure 1b appears to be oversaturated. Are the authors sure these images were measured within the analytical range of the microscope?

We set the same values for all the confocal images described in the paper. The expression of GFP-tagged MTA is much higher than other GFP-tagged m⁶A writer proteins and this is why the confocal images for it looks saturated while others are not (Fig. 2d). These saturated images has been replaced with new ones in which the smart gain value was decreased by 30% (Supplementary Fig. 1b).

- The authors should provide full names for all protein acronyms (e.g., HAKAI, MTA) when first mentioning those proteins.

Included as suggested

- The authors pose the scenario that the reduced level of m⁶A seen in the hakai lines could be due to a general reduction of methylation across all target transcripts, or from loss of methylation from certain proteins. While the authors provide their MeRIP-Seq results with GO enrichment analysis in this section, they should be more explicit with their conclusion of the reason behind this reduced level of m⁶A.

We have clarified this with the inclusion of the following text;

The distribution of m⁶A peaks was very similar between *hakai-2* and WT, and the number of peaks that were methylated in WT but which were absent or substantially reduced in *hakai-2* were fewer than those that were missing in the *fip37-4* and *vir-1* mutants. Thus, the majority of the 35% reduction in m⁶A abundance seen in *hakai* mutants is likely due to a global

reduction in methylation. However, a small number of transcripts appear to be disproportionately affected, and we found that those transcripts which are methylated in WT but not in *hakai-2* were enriched in association with root hair development processes.

- On page 11, lines 296-297: could the authors please clarify what they mean by "...and the number of peaks that were methylated in WT and disappeared or were substantially reduced in *hakai-2* were the lowest compared to the other two mutants" in the text?

We have clarified this point by changing the text as below;

The distribution of m⁶A peaks was very similar between *hakai-2* and WT, and the number of peaks that were methylated in WT but which were absent or substantially reduced in *hakai-2* were fewer than those that were missing in the *fip37-4* and *vir-1* mutants.

Reviewer 4

For HIZ1 I am curious about its expression during plant development. I wonder if the authors can check published RNA-seq data from different stages of development. I am curious what type of transcripts it tend to suppress methylation and functional implications of this suppression. I thought data mining and analysis can be quite useful for the community.

We have included additional root expression localization data (Supplemental Fig 4)

I would suggest the same for HIZ2. I wonder if the authors have at least RNA-seq data to compare HIZ2 mutant versus wt. A comparison could provide valuable information.

Additional MeRIPSeq analysis of *hiz2* mutants, under different environmental conditions, is something that we hope to undertake when time and resources allow.

A minor suggestion, in the introduction I would change fine tuning in "N⁶-methyladenosine (m⁶A) is necessary for the fine tuning of developmental and cell differentiation processes..." to regulating. M⁶A is quite important to mammalian development.

This sentence has been changed to;

N⁶-methyladenosine (m⁶A) is necessary for the regulation of developmental and cell differentiation processes in most eukaryotes...

Reviewers' Comments:

Reviewer #1:

Remarks to the Author:

All of my previous comments/concerns have been addressed in this revised form of the manuscript. I believe this study presents some very interesting new findings for this field of research.

Reviewer #2:

Remarks to the Author:

Although this revised manuscript is a bit improved, most of my major concerns are not addressed in this version as indicated below. Thus, I am unable to recommend its publication in a high impact journal like Nature Communications.

1. I have commented that the function of HIZ1 (in affecting m6A level and plant phenotypes) is mostly concluded from its overexpression phenotypes, which questions the endogenous function of HIZ1 in vivo. The authors did not directly answer this question. They argued that HIZ1 expression is restricted to a small number of cells. However, if indeed HIZ1 is important, it should at least influence the tissues/organs bearing these cells. If the authors want to conclude "HIZ1 is one of the novel proteins with opposing m6A functions." as clearly shown in the title, they should demonstrate more convincing data in addition to the overexpression results.

2. I asked about the targets of HIZ2 and the interaction between HIZ2 and other m6A writers. The authors only took advantage of Reviewer 1 comments to indicate that these experiments could be done in future studies. I disagree with this excuse. In this study, the authors have found an 85% decrease in m6A levels in hiz2 mutants, which also show strong developmental defects. Compared to hiz1, an in-depth characterization of hiz2 will clearly allow the authors to establish its function. Unfortunately, the authors only showed some phenotypic data on hiz2, but did not characterize further how HIZ2 is another novel protein that interacts with HAKAI.

3. I suggested the authors to show the concrete mechanism that links m6A writers and auxin response. In the revised manuscript, the authors showed decreased ARF19 and AUX1 expression and m6A modification in vir-1. However, it is unclear to me how this is relevant to the effects of HIZ1 and other known m6A writers.

4. I suggested the authors to show other pieces of evidence such as pull-down or CoIP assays to support their conclusion that HAKAI mediates the interaction between HIZ1 and MTA. However, they did not provide any further evidence in the revised manuscript. Since they identified HAKAI or MTA interacting proteins in different backgrounds by crosslinking the nuclear fractions with the formaldehyde solution (which can also fix protein-RNA or protein-DNA complexes), further direct evidence should be provided to confirm that HIZ1 indeed interacts with MTA in the presence of HAKAI. The existing data does not exclude the possibility that HIZ1 is only associated with MTA through HAKAI or other RNA/DNA molecules as scaffolds.

Reviewer #3:

Remarks to the Author:

Zhang and coauthors have presented a thorough analysis of two novel zinc finger proteins, HIZ1 and HIZ2, which have essential roles in the m6A writer complex and root development in Arabidopsis. The use of genetic phenotyping combined with fluorescence microscopy, immunoprecipitation, mass spectrometry, and other methods produce complementary evidence that paints a compelling picture in support of the authors conclusions that HIZ2 is a critical member of the Arabidopsis m6A writer

complex while HIZ1 is a HAKAI-dependent negative regulator of m6A deposition. This work is a significant step forward toward the understanding of mRNA methylation in Arabidopsis and will be well received by the community of associated plant scientists.

The authors have successfully integrated many reviewers' comments, including additional complementary experiments, that have further greatly improved the manuscript. The more stringent 5% and 1% FDR cutoffs applied show the target proteins are still present in the pull downs, which was a previous point of concern. The increased detail in methods will be useful for researchers seeking to reproduce the results. The authors have also added additional clarifying text throughout that has greatly increased the manuscript.

Reviewer #4:

Remarks to the Author:

These are quite interesting discoveries and phenotypes. I recommend publishing the revised manuscript.

Response to Reviewers' comments

Reviewer #1 (Remarks to the Author):

All of my previous comments/concerns have been addressed in this revised form of the manuscript. I believe this study presents some very interesting new findings for this field of research.

No additional action required

Reviewer #4 (Remarks to the Author):

These are quite interesting discoveries and phenotypes. I recommend publishing the revised manuscript.

No Additional action required

Reviewer #3 (Remarks to the Author):

Zhang and coauthors have presented a thorough analysis of two novel zinc finger proteins, HIZ1 and HIZ2, which have essential roles in the m6A writer complex and root development in Arabidopsis. The use of genetic phenotyping combined with fluorescence microscopy, immunoprecipitation, mass spectrometry, and other methods produce complementary evidence that paints a compelling picture in support of the authors conclusions that HIZ2 is a critical member of the Arabidopsis m6A writer complex while HIZ1 is a HAKAI-dependent negative regulator of m6A deposition. **This work is a significant step forward toward the understanding of mRNA methylation in Arabidopsis and will be well received by the community of associated plant scientists.**

The authors have successfully integrated many reviewers' comments, including additional complementary experiments, that have further greatly improved the manuscript. The more stringent 5% and 1% FDR cutoffs applied show the target proteins are still present in the pull downs, which was a previous point of concern. The increased detail in methods will be useful for researchers seeking to reproduce the results. The authors have also added additional clarifying text throughout that has greatly increased the manuscript.

No Additional action required

Reviewer #2 (Remarks to the Author):

Although this revised manuscript is a bit improved, most of my major concerns are not addressed in this version as indicated below. Thus, I am unable to recommend its publication in a high impact journal like Nature Communications.

1. I have commented that the function of HIZ1 (in affecting m6A level and plant phenotypes) is mostly concluded from its overexpression phenotypes, which questions the endogenous function of HIZ1 in vivo. The authors did not directly answer this question. They argued that HIZ1 expression is restricted to a small number of cells. However, if indeed HIZ1 is important, it should at least influence the tissues/organs bearing these cells. If the authors want to conclude "HIZ1 is one of the novel proteins with opposing m6A functions." as clearly shown in the title, they should demonstrate more convincing data in addition to the overexpression results.

2. I asked about the targets of HIZ2 and the interaction between HIZ2 and other m6A writers.

The authors only took advantage of Reviewer 1 comments to indicate that these experiments could be done in future studies. I disagree with this excuse. In this study, the authors have found an 85% decrease in m6A levels in *hiz2* mutants, which also show strong developmental defects. Compared to *hiz1*, an in-depth characterization of *hiz2* will clearly allow the authors to establish its function. Unfortunately, the authors only showed some phenotypic data on *hiz2*, but did not characterize further how HIZ2 is another novel protein that interacts with HAKAI.

3. I suggested the authors to show the concrete mechanism that links m6A writers and auxin response. In the revised manuscript, the authors showed decreased ARF19 and AUX1 expression and m6A modification in *vir-1*. However, it is unclear to me how this is relevant to the effects of HIZ1 and other known m6A writers.

4. I suggested the authors to show other pieces of evidence such as pull-down or CoIP assays to support their conclusion that HAKAI mediates the interaction between HIZ1 and MTA. However, they did not provide any further evidence in the revised manuscript. Since they identified HAKAI or MTA interacting proteins in different backgrounds by crosslinking the nuclear fractions with the formaldehyde solution (which can also fix protein-RNA or protein-DNA complexes), further direct evidence should be provided to confirm that HIZ1 indeed interacts with MTA in the presence of HAKAI. The existing data does not exclude the possibility that HIZ1 is only associated with MTA through HAKAI or other RNA/DNA molecules as scaffolds.

1) in line with this referee and the Editor's suggestions we have changed the title of the manuscript to **“Two zinc finger proteins with functions in m⁶A writing interact with HAKAI”**

2) Whilst the molecular function of HIZ2 within the complex remains unclear, this is also the case for the presumed human counterpart ZC3H13. Indeed, the essential functions of the core m6A writer complex members, VIR/VIRMA and HAKAI/CBLL1, of both metazoans and plants remains unclear despite their presence in the writer complex being known for some years. The finding that a plant orthologue of ZC3H13/FLACC is indeed present in the plant complex (despite earlier reports that plants lacked this component) is highly significant and will be of broad interest. However, deeper understanding of HIZ2/ZC3H13 molecular function, as well as the other conserved complex members, will require in depth molecular, biochemical and structural studies.

3) As this reviewer indicates, in the previous revision, we addressed possible roles of m⁶A in auxin responses (in answer to reviewer 1's previous questions). We showed the *vir* data for *arf19* and *aux1* as the *vir* mutant has the lowest m6A levels (and most severe auxin phenotypes) of the different writer complex members.

4) The experiments described in the results section are CoIP experiments. Both MTA and HAKAI bring down HIZ1. Furthermore, HIZ1 is no longer brought down by MTA in the absence of HAKAI. In addition, the genetic experiments in which HIZ1 OE phenotypes are lost upon crossing to the *hakai* mutant background further support this as a functional interaction.